# CMOS Interfaces for Internet-of-Wearables Electrochemical Sensors: Trends and Challenges

**Michele Dei [1],\***  , **Joan Aymerich [1]**, **Massimo Piotto [2]**  , **Paolo Bruschi [2]**  ,
**Francisco Javier del Campo [1]**  and **Francesc Serra-Graells [1,3]**  

[1]   Institut de Microelectrònica de Barcelona IMB-CNM (CSIC), 08193 Barcelona, Spain;
joan.aymerich@imb-cnm.csic.es (J.A.); javier.delcampo@csic.es (F.J.d.C.);
paco.serra@imb-cnm.csic.es (F.S.-G.)

[2]   Dipartamento di Ingegneria dell'Informazione of University of Pisa, 56126 Pisa, Italy;
massimo.piotto@unipi.it (M.P.); paolo.bruschi@unipi.it (P.B.)

[3]   Department of Microelectronics and Electronic Systems, Universitat Autònoma de Barcelona,
08193 Barcelona, Spain

\*   Correspondence: michele.dei@imb-cnm.csic.es; Tel.: +34-935-94-77-00 (ext. 2492)

**Abstract:** Smart wearables, among immediate future IoT devices, are creating a huge and fast growing market that will encompass all of the next decade by merging the user with the Cloud in a easy and natural way. Biological fluids, such as sweat, tears, saliva and urine offer the possibility to access molecular-level dynamics of the body in a non-invasive way and in real time, disclosing a wide range of applications: from sports tracking to military enhancement, from healthcare to safety at work, from body hacking to augmented social interactions. The term *Internet of Wearables* (IoW) is coined here to describe IoT devices composed by flexible smart transducers conformed around the human body and able to communicate wirelessly. In addition the biochemical transducer, an IoW-ready sensor must include a paired electronic interface, which should implement specific stimulation/acquisition cycles while being extremely compact and drain power in the microwatts range. Development of an effective readout interface is a key element for the success of an IoW device and application. This review focuses on the latest efforts in the field of Complementary Metal–Oxide–Semiconductor (CMOS) interfaces for electrochemical sensors, and analyses them under the light of the challenges of the IoW: cost, portability, integrability and connectivity.

**Keywords:** integrated circuits design; smart wearables; Internet of Wearables; Internet of Things; CMOS Electrochemical Sensing; Flexible Technologies; Hybrid Integration Technologies

## 1. Introduction

At the onset of the era of connected intelligence, many real-life issues have been addressed through massive data collection from a community of users. Internet-of-Things (IoT) devices are already shaping our lives through augmented information, entertainment and social interactions. Recently, the IoT market has been experiencing the exponential diffusion of smart unobtrusive wearable devices aimed at determining important vital parameters of human beings. In this paper, we will discuss the term Internet of Wearables, IoW, for these extremely innovative devices distinguished from the previous generation of portable biomedical devices for their miniaturization, portability and internet-ready communication capabilities. In this scenario, miniaturized electrochemical sensing are likely to play a key role in many contexts including: military, security, food quality and safety, healthcare, wellness and environmental monitoring. The relevance of electrochemical sensing through dedicated CMOS interfaces is demonstrated by a number of recent reviews [1–3]. The ambition of this

work is to provide a complete survey giving all the relevant information about the electrochemical sensing and the most advanced CMOS instrumentation circuits and techniques with an emphasis on Internet-of-Wearables (IoW) application. This article is organized as follows: Section 2 gives an overview of the latest trends in smart wearables, Section 3 summarizes the fundamentals of electrochemical sensing most commonly used in the field of wearables, Section 4 offers an extensive and detailed account of CMOS interfaces for electrochemical sensors. This work concludes by discussing the main challenges facing CMOS design for IoW.

## 2. Smart Wearables and the IoT: Basic Requirements and Main Fields of Application

Wearable technologies are one of the fastest growing industries of our time, expected to generate revenues over $51 bn by 2022 at a gaping compound annual growth rate of 15% [4]. Smart wearables are electronic devices whose mission is to interface humans with the digital world [5]. The goal is to improve the quality of life or performance of the users by sensing the person wearing it and its environment, and providing these data to a processing unit able to inform or assist the user in different ways depending on the specific application.

Pioneering devices, such as activity monitors, successfully integrate multiple physical sensors. However, without complementary biochemical information, it is nearly impossible to form a complete picture of an individual's health status in real time. However, the integration of chemical sensors is more challenging compared to physical ones due to the differences between them in terms of fabrication technologies, calibration requirements, and operating lifetime. As will be discussed below, physical sensors are in many cases compatible with CMOS processes and many conventional fabrication techniques, and have lifetimes on the order of months or years. Chemical sensors, on the other hand, are incompatible with many microelectronic processes, particularly CMOS, and require frequent calibration or replacement. Consequently, huge opportunities are emerging in the area of chemical sensing, but they come hand in hand with no less significant challenges [6].

Nowadays, the development of smart wearable chemical sensors is driven by applications in four key areas: (i) health; (ii) sports; (iii) work safety; and (iv) defense and law enforcement. There are a number of analytes of common interest across these application areas, such as glucose and electrolytes, and then there are other analytes of particular relevance to a single application, as in the case of certain hormones, drugs of abuse and toxins. Regarding transduction mechanisms, most sensors rely on either optical or electrochemical sensors. This review addresses recent development in CMOS interfaces for the latter type.

### 2.1. Wearables and Non-Invasive Monitoring

Figure 1 depicts suitable body areas for placement of wearable chemical sensors, as well as typical form factors and application areas. On the other hand, Table 1 provides physiological concentration ranges of analytes of interest in wearable chemical sensors in body fluids suitable for the development of non-invasive detection, including wearable technologies [7]. Ideally, wearable chemical sensors should be non-invasive and, among candidate body fluids, such as sweat [6], saliva [8,9], urine and tears [10], sweat seems the most suitable medium to address biochemical parameters unobtrusively. This is reflected by the relatively higher number of works reporting sweat sensors [11–15] in the literature, compared to saliva or tears. Urine is a special case because, although it enables non-invasive determination of biochemical parameters, it does not lend itself for non-invasive monitoring applications. Depending on which of these fluids is addressed, the most common sensor forms are skin-patches, followed by contact lenses. Salivary analysis, on the other hand, is typically performed from swabs.

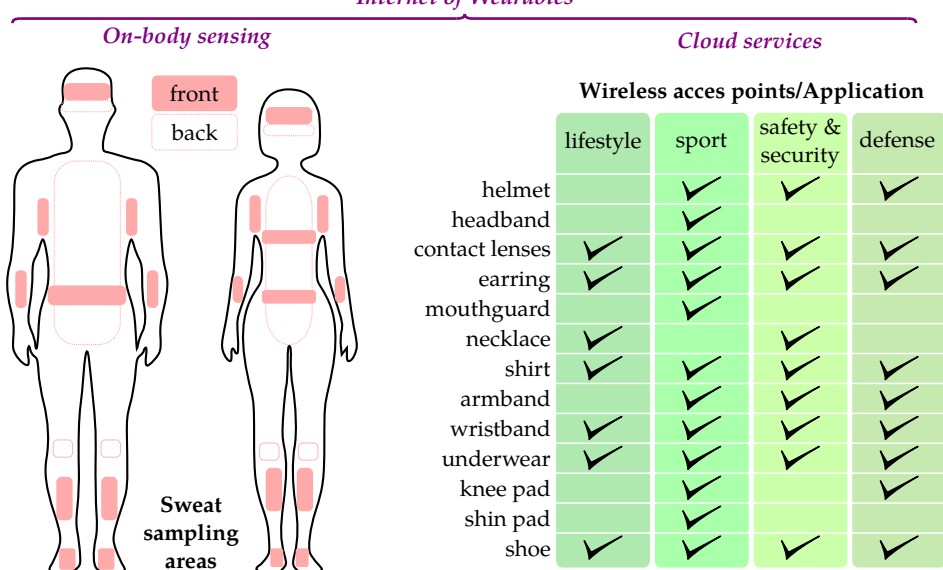

**Figure 1.** Internet of Wearables concept. Sweat sampling areas in the human body for in situ measurement of physiological analytes and possible wireless access points for data links.

**Table 1.** Normal ranges for analytes of interest in body fluids.

| Body Fluids | Blood | Sweat | Saliva | Tear | Urine [a] |
|---|---|---|---|---|---|
| **References** | [16] | [17–20] | [8,9] | [16,21–23] | [24–27] |
| **Glucose [mM]** | 3.3–6.7 | 0.06–0.11 | 0.22–0.72 | 0.2–1 | 2.78–5.5 |
| **pH** | 7.36–7.44 | 4–5.5–7 | 6.2–7.6 | 6.5–7.6 7.14–7.82 | 4–8 |
| **[Na$^+$] [mM]** | 136–145 | 10–40 [b] | 20–80 | 80–161 | 40–220 |
| **[K$^+$] [mM]** | 3.5–5 | 4–5 [c] | 20 | not found | 25–125 |
| **[Cl$^-$] [mM]** | 98–107 | <40 | 30–100 | 106–130 | 110–250 |
| **Lactate [mM]** | 0.3–1.3 [d] 0.36–0.75 [e] | 20–60 5–110 | 0–0.4 [f] | unknown | 0.5–2.2 [g] |
| **IL-6 [pG/mL]** | 0–4.3 | 7–16 | 2.5 | 100–200 | 20–30 |

[a] For urine, quantities are expressed as [mmol/day]; [b] 70 mM suggests cyctic fibrosis; [c] [K$^+$] > 10 mM indicates a likely system error; [d] venous blood; [e] arterial blood. During exercise/extreme exercise up to 12–25 mM; [f] values during/after physical exercise; [g] Measured in the interval between the first morning and the previous urine sample; first morning urine sample was obtained between 6:00 a.m. and 10:00 a.m.

## 2.2. Wearable Sensors in Health

The pursuit of chemical sensors for non-invasive monitoring has been a truly active field of research over the past few decades [28]. Wearable sensors can be of great help in the early diagnosis and prevention of various diseases, particularly those associated with lifestyle such as diabetes mellitus and cardiovascular diseases. Its growing prevalence makes diabetes mellitus [29] perhaps the single most important application of wearable chemical sensors today. Blood glucose is measured using amperometric or coulometric biosensors and is able to quantify the oxidation of glucose using oxidase [30] or dehydrogenase enzymes [31]. While enzyme-based systems dominate, long-term stability issues associated with enzymes have motivated very active research in non-enzymatic glucose sensors [32]. Additionally, non-enzymatic glucose sensors might be easier to produce than their enzymatic counterparts, which will eventually represent an important breakthrough.

The evolution of glucose biosensors has been addressed in countless articles and excellent reviews [23,31], including wearable and non-invasive ones [33], but a summary will be given for

convenience. Glucose monitors have evolved greatly [23,34] since Clark and Lyons published the first glucose biosensor back in the 1960s [35]. Blood glucose analyzers have evolved from laboratory benchtop instruments, first commercialized by Yellow Springs Instruments (YSI Inc., Yellow Springs, OH, USA) in 1975, to home appliance in the 1980's, to portable, hand-held self-monitors in the 1990s. These instruments analyze blood using enzyme biosensors, and although they have been able to work on increasingly smaller blood samples (most current systems require less than 1 μL of blood), they still require uncomfortable and painful blood extraction using a lancet. An important technology leap was made by Cygnus' (Cygnus Inc., Redwood City, CA, USA) Glucowatch in the late 1990s [36]. Glucowatch relied on two glucose biosensors to monitor glucose in sweat, following stimulation by reverse iontophoresis through pilocarpine pads. Unfortunately, issues related to calibration, quantification of sweat glucose, and skin irritation meant the end of the Glucowatch, but it was nevertheless a very significant milestone in terms of technology and product development. Since then, the industry has gone back to an intermediate stage, between invasive daily finger-pricking and non-invasive sweat measurements, to monitor blood glucose through minimally invasive monitoring in interstitial fluid. Abbott's (Abbott Laboratories, Chicago, IL, USA) Freestyle Libre [37] monitors blood glucose through interstitial fluid over a period of two weeks [38], which represents a major improvement for the lifestyle of diabetic patients.

### 2.3. Wearable Sensors in Sport

Chemical sensors in sport wearables are intended to ensure top-performance in athletes [39]. This may be achieved through injury prevention and early detection [40], but also through the monitoring of effort and hydration in real time through sweat, which is the obvious sample medium to monitor, given its abundance during physical exertion [41]. In spite of this, sweat measurements also require choosing an adequate body area for sampling [17] and smart sweat management strategies [42], the most common of which are based on microfluidics [43–45] or on absorbents and lateral flow membranes [15,20,46].

The prevention of injuries, particularly in elite and professional sports, is an ambitious and elusive goal. One way to achieve it is through the early detection of biomarkers of muscular damage.

Two examples of such biomarkers are creatine kinase and cytokines such as interleukin-6 (IL-6) [47]. While creatine kinase does not show in sweat, IL-6 has been reported to appear in sweat at similar concentration levels as blood [48]. Its determination requires the use of affinity-based sensors, which, in contrast to enzyme-based biosensors, are generally unsuitable for monitoring purposes. Aptasensors, a type of affinity-based sensor that relies on synthetic receptors [49], are more reversible than immunosensors and may be adapted to semi-continuous measurements. An example of this has been recently reported of label-less detection of IL-6 in artificial sweat using electrochemical impedance spectroscopy [50]. Perhaps the most accessible analytes in sweat are electrolytes, which provide direct information on hydration levels, and are measured potentiometrically using ion selective electrodes [51]. Dehydration is an important condition to determine and manage, as total body water losses greater than 2% are likely to cause performance losses in endurance sports [52]. Other analytes, such as glucose and lactate, may also be determined in sweat, using amperometric biosensors instead. However, although the correlation between sweat and blood glucose is accepted, the case of lactate, an indicator of muscular effort, is not so clear-cut. Blood lactate is typically measured in capillary blood, typically from the ear lobe [53]. Although the correlation between blood lactate and sweat lactate levels has been reported [19], and it is accepted that sweat lactate may be used as an indirect indicator of physical activity, sweat lactate is a more likely indicator of sweat gland activity itself [54].

### 2.4. Safety at Work

This is another area where non-invasive monitoring of biochemical parameters can help prevent accidents and injuries in physical work environments. Examples of wearables featuring physical sensors can be found in the literature [55].

In terms of chemical and biochemical information, the same parameters and sensors applied in sport could be applied to the workplace to monitor the fitness level of a pool of workers. In addition to level of hydration and effort, other risk factors such as stress and drugs of abuse may be detected non-invasively too [56]. Small sized hormones, such as serotonine, dopamine and cortisol, may be used to assess an individual's emotional state non-invasively, as they appear in sweat, saliva, tears and urine [27]. The difficulties in the detection lie in the low concentration levels of these molecules in sweat (in the nM-μM range), and the fact that detection in most cases requires immunosensors, which, as mentioned earlier, are unsuitable for continuous measurements required in a monitoring application. Barring ethanol, which is easily detected in breath and sweat electrochemically, the intake of most drugs of abuse is very hard to monitor non-invasively and impractical to embed in a wearable device because their detection involves immunoassays.

### 2.5. Defense and Law Enforcement

This is another area of great interest, which can also benefit from the same technologies described above for sports. However, these applications normally have much more stringent requirements in terms of reliability and robustness. This makes them particularly challenging to develop, but, correspondingly, highly valuable and attractive in economic terms. Recent examples in this area involve wearable sensors for the detection of nerve agents [57–59].

### 2.6. Design and Fabrication of Wearable Devices

Application scenarios define the functional requirements of wearable devices. This section aims to highlight a few important ideas that need to be considered in relation to the design of wearable sensors. The form factor of a wearable device defines aspects such as its physical parts, their shape and size. Most wearable devices known today, such as activity monitors and smartwatches, are "one-part" devices because they rely on physical sensors alone. This is possible because those physical sensors will last as long as the device itself—whose lifetime is usually limited by its power source—and because they are produced using Microelectromechanical systems (MEMS) technologies that are fully or highly compatible with the CMOS processes employed in the fabrication of microprocessing units. Therefore, it is relatively easy to integrate them inside or along other silicon components, relying on System-on-Chip, SoC, technologies vs. System-in-package, SiP, and heterogeneous integration processes [60,61].

Chemical sensors, on the other hand, are much shorter lived than any other component in these smart microsystems. The (bio)sensor recognition element determines the longevity. Thus, ignoring calibration, ion selective electrodes may operate for weeks, enzyme biosensors can last hours to days, and affinity-based biosensors, such as immunosensors, may just be good for a single use or a few hours. Although different approaches have been taken to overcome their operational stability problems, the most common one consists in separating the sensor from the electronics, as in the case of present-day glucometers, where the biosensing part is a disposable strip that is inserted inside a reader. The same model was applied to the Glucowatch, and only Abbott's Freestyle Libre departs from it by integrating the biosensor into a disposable microsystem that interfaces wirelessly with the reader. In this last case, there is a real advantage if the benefits for the user outweigh the (economic and environmental) costs of disposing of a button battery, a printed circuit board, PCB, containing several discrete components, and a plastic casing. This approach of integrating discrete components in a disposable package has also been adopted in several works present in the literature. However, even if the components can be produced inexpensively, the cost of heterogeneously integrating them in flexible or elastic substrates, is still too high [62–64].

We can see only two solutions to this problem until the cost of integration becomes affordable. One is to reduce the overall number of silicon components to a single chip, and with as few contact pads as possible. The other solution involves devices that do not require silicon or any kind of discrete

components at all [65]. This second approach may have been completely unfeasible only a few years ago, but progress in printed electronics may soon enough make it possible [66–68].

In both cases, interfaces between the wearable and the user and between the wearable and the rest of the world needs to be carefully studied. The first one defines important aspects of usability and comfort, while the second set of interfaces determines how a disposable sensor interacts with its control instrumentation. The physical connection required between electrochemical sensors and their control instrumentation, namely a potentiostat, represents a significant hurdle in a wearable system. In order to avoid wiring and connectors, which would increment the complexity and the cost of heterogenous solutions, recent works focused on integration of the CMOS die along sensor and data-linked wirelessly via Near-Field Communication (NFC) or other short-range radio-frequency (RF) signals [14,69,70]. Works reporting wireless links are still a minority, but they are growing rapidly [71]. RF can and should be used not only to extract information from the device, but also to power it. This can alleviate the need for on-board power sources and their associated control components, which take up precious space and can make the final device not only bulkier, but also much more costly.

### 2.7. Data-Access Points: Smartphone, Smartwatch and WPAN Radio

The next generation of smart wearables will adhere to the IoT paradigm embodying user-centred Cloud-connectedness devices. The user will seek control and interaction with data generated from its body and/or its surroundings without the burden of heavy and bulky devices. Connection to the Cloud is also fundamental to enrich the wearable experience through ambient intelligence. In order to emphasize this latter concept, we coin here the term Internet of Wearables. The IoW concept ecompasses a number of aspects which go beyond the scope of this review. Nevertheless, we find it useful to employ this term since it entails specific characteristics for the IoW interface circuits as will be clear in the following discussion.

Smartphones currently play a crucial role in the definition of the smart wearables, since they are the most popular and widespread IoT platforms. This has been the subject of a number of dedicated reviews due to their amenability as portable instrument for biosensing and self-diagnostics [72–75]. The importance of smartphones resides in the combination of the following factors: (i) they already carry powerful data processing hardware; (ii) they enable direct and intuitive user interfacing; (iii) they allow for Cloud connectivity and (iv) they provide a number of data-access ports including camera, USB, NFC and audio-jack that can be exploited to interface with the smart wearables.

Smartwatches are also gaining popularity as powerful e-gadgets and have also been a subject of interest for researchers as a tool to gather biometric data [76,77]. Their body conformability and low weight represent an important step ahead towards the IoW.

IoW devices may communicate each other directly through medium/short-range wireless connectivity or through another device, e.g., a smartphone. In both cases, radio capabilities must be embedded. In order to enable interoperability between the IoW devices and their surroundings, compliance with wireless standards is a compulsory need. In this sense, a number of standards from the IEEE 802.15 working group and from the International Organization for Standardization are of interest for IoW applications.

Table 2 lists the trends in the state-of-the-art research on smart wearables equipped wireless off-the-shelf components. It emerges that wireless personal area networks, WPANs, such as Zigbee and Bluetooth are currently employed for medium/short range communications. The Bluetooth Low Energy variant, is particularly attractive for its smart power management that allows for a long battery life. An IoW-ready CMOS interface must be capable of communicating with the transceiver chip through the provided digital I/O interfaces, usually comprising Serial Peripheral Interface (SPI) and/or I2C protocols.

For totally passive, i.e., battery-less, operation, the NFC technology is employed. In this case, the IoW interface must include all the RF front-end circuits (rectifier, load modulator, limiter, low-dropout regulator, envelope detector, demodulator and clock recovery circuit) and the digital

control logic needed to implement the specific NFC ISO/IEC protocol used for the application. The great advantage of getting rid of the burden of the battery comes at the cost of limited wearability, since the smartphone or other NFC-capable device needs to be kept aligned with the sensing tag during the measurement cycles, which, in the case for electro-chemical sensing, may extend in the range of minutes.

New possibilities may emerge in the immediate future from research in bio-fuel cell sensing, where both analytical information and operational power can be extracted from the electrochemical interface. Recently, authors in [78] demonstrated that the power harvested from an electronic skin bio-fuel cell is high enough to operate a conventional Bluetooth Low Energy (BLE) radio chip. Section 4.5 reports the last pioneering custom integrated circuits interfaces in this field.

**Table 2.** Wireless personal area networks-radio options based on off-the-shelf components for Internet of Wearables.

| Radio | Off-The-Shelf Component(s): Radio + MCU | Reference(s) |
|---|---|---|
| Zigbee (IEEE 802.15.4) | Xbee-PRO RF + PIC24HJ12GP201 | [79] |
| | Xbee + Arduino Lilypad | [80] |
| | nRF2401 + ATMega128L | [81] |
| | CC2530 | [82] |
| Bluetooth Low Energy (IEEE 802.15.1) | CC2541 | [83–85] |
| | HC-06 + ATMega328p | [43,64,86] |
| Bluetooth 2.0 | CC2560 + MSP430BT5190 | [87] |
| NFC/RFID (ISO 15693) | Commercial tag, custom modified | [88] |
| | SL13A | [89] |
| | MLX90129 | [62,90] |

## 3. Electrochemical Sensing Overview

Electroanalytical methods provide specific information on chemical and biochemical systems by looking at the interplay between chemical species and electrical quantities at the interface between the sensor and the test solution.

When an electrode is immersed in a solution, a charge separation occurs at the interface, and an electrical double layer is established. The charge developed at the electrode surface is balanced by the solution across a small volume consisting of both adsorbed ions and ions in solution. Overall, this volume extends over a distance of a few nanometers. This is where (most of) the electrode potential is developed and electron transfer between the electrode and species in solution takes place [91,92].

The electrochemical nature of the electrode-solution interface can be used to develop (electro)analytical methods reliant on the relation between electric current, potential or impedance and the concentration of a target analyte [93]. Thus, electroanalytical methods are classified according to the measured electrical quantity as: (i) potentiometric; (ii) amperometric; and (iii) impedance-based.

### 3.1. Potentiometric Sensing

The impossibility to measure the potential drop across a single electrode–solution interface determines that two electrodes are required to carry out potentiometric measurements.

As depicted in Figure 2, these two electrodes are: an indicator electrode, i.e., an ion-selective electrode (ISE), and a reference electrode, RE, whose potential is stable and independent of the sample composition. In order to ensure the stability of the reference electrode potential, an important feature of potentiometric measurements is that they are carried out on an open circuit. Thus, any potential change

measured across this two-electrode cell can be exclusively attributed to the indicator electrode–solution interface, and related to analyte concentration changes through an expression of the kind:

$$V_{cell} = constant + \frac{RT}{nF} \ln [analyte]. \tag{1}$$

In Equation (1), $R$ is the universal gas constant, $T$ is the temperature, $F$ is the Faraday's constant and $n$ is the valency of the ion. In the following discussion, we will use the known equivalence expression containing the Boltzmann constant $K$ and the fundamental electron charge $q$:

$$\frac{RT}{F} = \frac{KT}{q} = U_T, \tag{2}$$

where $U_T$ is the thermal voltage, according to a definition inherited from the semiconductor physics terminology.

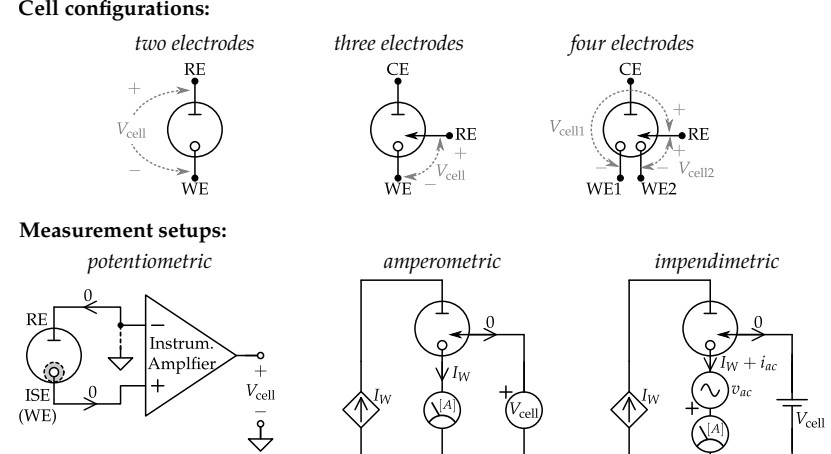

**Figure 2.** Electrochemical transducer cell typical configurations and measurement setups.

In an ion-selective electrode [94], the indicator electrode presents a membrane containing an ionophore that is selective to the target ion. However, in practice, other ions may also permeate through the membrane and affect the sensor response.

The typical response at an ion selective electrode is depicted in Figure 3. At low analyte concentrations, the sensor response is determined by the concentration of the interfering ions. Above a certain concentration, i.e., the limit of detection, the sensor response is linearly dependent on the logarithm of the analyte concentration. In the case of singly charged species, a slope of $U_T / \log_{10} e \approx 59$ mV per decade (of concentration at room temperature) is referred to as Nernstian, and it can be used to make a quick estimate of the resolution and sensitivity required in the instrumentation for a particular application.

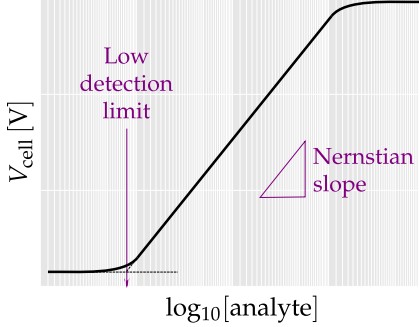

**Figure 3.** Schematic Ion-Selective electrode (ISE) static characteristic.

### 3.2. Amperometric Sensing

Amperometric techniques are based on the measurement of the current response at a given electrode potential. In contrast to potentiometric sensors, which only require two electrodes, amperometric techniques require three electrodes, namely working (sensor), reference, and auxiliary electrodes, indicated in Figure 2 as WE, RE and CE, respectively. The role of the RE in amperometry is to control the potential at the WE. To enable this control, no current can flow through the RE, and hence the need for a third, auxiliary electrode whose function is to supply the current required by the WE, $I_W$. Since the CE is typically larger than the WE, the process occurring at CE does not limit $I_W$.

Electrode processes are generally represented by an equation such as:

$$O + n\text{e} \underset{k_b}{\overset{k_f}{\rightleftharpoons}} R, \tag{3}$$

where O and R are the respective oxidized and the reduced form of a species, e is the transferred electron and $k_f$ and $k_b$ are the reduction and oxidation rates, respectively. Equation (3) describes a general redox process, which is also referred to faradaic process, since it is governed by Faraday's law stating that the amount of chemical reaction (mass) caused by the flow of current is proportional to the total charge passed through the electrode/solution interface. Rate constants $k_f$, $k_b$ are potential dependent:

$$k_f(V_{\text{cell}}) = k_0 \exp\left(-\alpha \frac{V_{\text{cell}} - U_0}{U_T}\right); \qquad k_b(V_{\text{cell}}) = k_0 \exp\left((1-\alpha)\frac{V_{\text{cell}} - U_0}{U_T}\right). \tag{4}$$

When $V_{\text{cell}}$ equals the standard potential $U_0$, both $k_f$, $k_b$ are equal to $k_0$ which is known as the standard heterogeneous rate constant. The standard potential, $U_0$, defines the activation threshold of the redox process, while $k_0$ measures the kinetic facility of the redox couple. The charge transfer coefficient $\alpha$ indicates the symmetry of the electron-transfer process. It is commonly assumed to be around 0.5, although in most cases it is found to be between 0.3 and 0.7. The $k_f$, $k_b$ rate constants are controlled by the voltage drop across the electrical double layer, which in turn depends on the potential applied between the working electrode, WE, and the RE.

Notation in Equation (4) differs slightly from classical notations used in Electrochemistry books with the purpose to highlight similarities with the bipolar transistor and diode models which are more familiar in the field of electronic engineering.

The faradaic process is then governed by the current-overpotential equation:

$$I_{W,\text{faradaic}} = i_0 \left( \frac{c_{O,S}}{c_O^*} \exp\left(-\alpha \frac{V_{\text{cell}} - U_0}{U_T}\right) - \frac{c_{R,S}}{c_R^*} \exp\left((1-\alpha)\frac{V_{\text{cell}} - U_0}{U_T}\right) \right),$$
$$i_0 = FAk_0(c_O^*)^{(1-\alpha)}(c_R^*)^{\alpha}, \tag{5}$$

where $c_{O,S}$ and $c_{R,S}$ are the O and R concentrations at the WE, respectively, while $c_O^*$ and $c_R^*$ indicate the same concentrations in the bulk (far from WE). In dynamic conditions, $c_{O,S}$ and $c_{R,S}$ differ from their respective values in bulk and obey Fick's second law of diffusion. While analytical treatment is out of scope in the context of this review, details can be found in general Electrochemistry books [91,92]. However, Equation (5) already carries all the valuable information for sensing purposes: $i_0$ depends on the concentration of a target analyte, which may be whether O or R. Dynamic experiments, which involve temporal transients, also holds electro-analytical information through the $c_S/c^*$ ratios, which will be apparent in the following discussion.

Oxidation currents mean that electrons pass from the solution to the anode and, conversely, reduction currents involve the passage of electrons from the cathode to the solution. The CE completes the current path. At the same time, the current is sustained by ions in the electrolyte: in this region, the voltage drop between the working and the reference electrode, also known as ohmic, or *iR*-drop, should be minimized to ensure control over the working potential.

The current passing through the working electrode corresponds to the oxidation or reduction of the target analyte at the measurement potential. This measurement potential is chosen so that the current is not limited by the electrode kinetics, but mass-transport controlled. Analyte mass transport to the WE is influenced by diffusion, convection and migration. Amperometric experiments are usually designed for diffusion to prevail over the other transport mechanisms. In this way, temperature/density gradients and electrostatic forces do not influence $I_W$. Mass-transport control requires the application of a certain overpotential, which is the difference between the applied potential and the formal potential of the analyte redox process, $V_{cell} - U'_0$. The formal potential $U'_0$ differs from the previously defined standard potential $U_0$ in that $U'_0$ takes into account non-standard conditions, i.e., $T \neq 298\,\text{K}$, $\text{pH} \neq 0$, $[\text{analyte}] \neq 1\,\text{M}$ and how the medium or the presence of other electrolytes affects the activity of the analyte. While the applied potential should ensure mass transport control, it also ought to be low enough to prevent the oxidation or reduction of other electroactive species present in solution, and which may result in determination errors.

Controlled $V_{cell}$ potential electroanalytical methods include potential step and linear sweep-based techniques, such as cyclic voltammetry [91–93]. A key aspect of these techniques is that the initial potential must be such that no overall faradaic current is observed. In other words, the initial potential should be close to the open circuit potential. This facilitates that the current measured during the experiment is due to the main process of interest alone. Figure 4 represents the potential functions and current response corresponding to a potential step as well a cyclic voltammogram. In a simple potential step experiment, the transient current at the WE is monitored while $V_{cell}$ is stepped to the measurement potential, as sketched in Figure 4a. During the first part of the experiment, the current corresponds to the charging or discharging of the electrical double layer. Depending on the electrode size, material and structure, this charging current may last for up to several milliseconds, during which the underlying faradaic current is much weaker and hard to detect. Once the double layer is discharged, the remaining current corresponds to faradaic processes, and can be measured more safely. Advanced potential step techniques, such as square wave and differential pulse voltammetry, are designed to eliminate the charging current contribution and enhance the faradaic component, but they require relatively fast instrumentation and therefore their application in wearables is not very attractive.

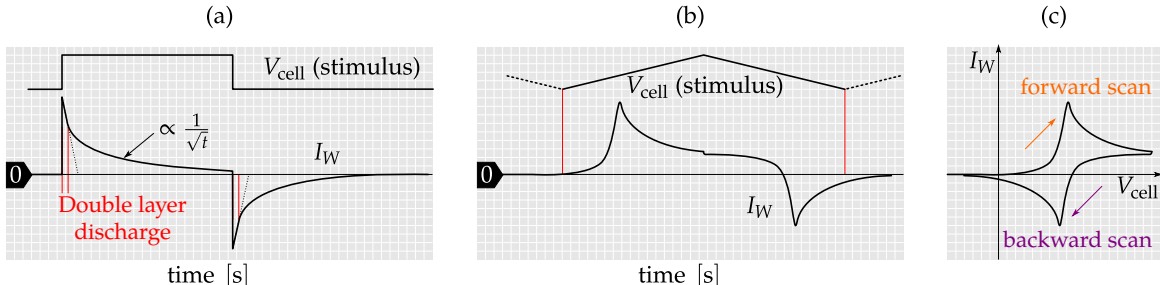

**Figure 4.** Electrochemical experiments: (**a**) chronoamperometry (versus potential step); (**b**) chronoamperometry (versus potential scan); (**c**) voltammogram.

Another well-established electrochemical method is cyclic voltammetry. In this case, a potential ramp is applied to the WE instead of a step, and the current response is recorded, as indicated in Figure 4b. At some point in the potential scale, well past $U'_0$, the scan direction is reversed. This results in the recording of a forward and a backward current scan (Figure 4c). Analysis of these current responses provides rich and extremely useful information. Peak currents and positions may be used to establish kinetic and thermodynamic information about the system at hand, and the scan rate provides access to additional information in the time domain.

Although most amperometric systems are based on three-electrode cells, occasionally we may also find four-electrode detection systems based on two working electrodes that can be independently biased. This is also schematized in Figure 2. These systems may be used either for the detection of two

independent analytes or in signal amplification strategies known as generator–collector systems [95–98]. In these systems, which require the analyte to be electrochemically reversible, an electrode is biased above and the other below the formal potential of the analyte. Thus, the oxidized and reduced species shuttle between electrodes, enhancing the measured current. Additionally, because many common analytes are irreversibly oxidized or reduced, using a redox cycling approach minimizes the contribution of these interfering species which cannot be regenerated [99].

### 3.3. Electrochemical Impedance Spectroscopy

Potentiometry and amperometry are direct current, DC, techniques. Electrochemical impedance spectroscopy, EIS, on the other hand, is based on the application of alternate signal, typically potential, across the working electrode [100]. The corresponding current response allows the calculation of the system impedance, which is a complex quantity with a real and an imaginary part. The most common way to analyze electrochemical impedance spectra is through the use of so-called equivalent circuits. An equivalent circuit is just a representation of the electrochemical system in terms of simple electric components such as resistors, capacitors and, in the presence of chemical reactions, even inductances [101].

The Randles circuit, depicted in Figure 5, is a common way to model a simple electrochemical system. The electrode–solution interface is represented by a capacitor, $C_{dl}$, representing the electrical double layer, in parallel with a second branch that accounts for a faradaic process. Such faradaic process consists of two components: a resistor, $R_{ct}$, which represents the charge transfer process, and a Warburg element $Z_w$ that accounts for mass transport of the electroactive species from the solution bulk to the electrode surface. Next, in series with this parallel branch, we find a resistor, $R_\Omega$, which represents the solution resistance between the WE and the RE. A similar circuit can be drawn for the auxiliary electrode, but since the WE processes are the limiting ones, the auxiliary electrode contribution is usually neglected. In addition, as will be discussed in the following sections, the feedback loop built driving the current at CE allows applying a known alternate current (AC) voltage between the RE and the WE, regardless of the impedance seen at the CE. This feedback loop has been indicated in the measurement setups of Figure 2 as a current-controlled generator, whose controlling quantity is the WE current, $I_W$.

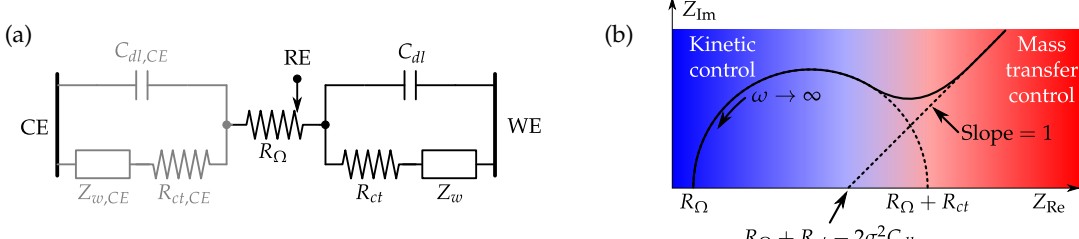

**Figure 5.** Randles equivalent circuit and Nyquist plot of electrochemical cell impedance seen between Counter electrode (CE) and working electrode (WE): (**a**) schematic; (**b**) characteristic on the real-imaginary plane.

In contrast with amperometric systems, where the electrode is driven to a condition far from equilibrium by the application of potential steps or potential sweeps, the EIS approach studies the system in a condition of quasi equilibrium. To this purpose, a small signal $v_{ac}$ perturbation (usually <10 mV) is applied around a fixed $V_{cell}$, as depicted in Figure 2. The steady-state condition of the electrochemical cell allows for the linearisation of Equation (5) which leads to the expression of $R_{ct}$

$$R_{ct} = \frac{U_T}{ni_0}. \tag{6}$$

The Warburg impedance $Z_w$ has a frequency ($f$) dependence law, and is commonly expressed as

$$Z_w(\omega) = \frac{\sigma}{\sqrt{\omega}}(1 - \jmath); \quad \sigma = \frac{U_T}{n^2 FA\sqrt{2}}\left(\frac{1}{\sqrt{D_O}\,c_O^*} + \frac{1}{\sqrt{D_R}\,c_R^*}\right), \tag{7}$$

where $\omega = 2\pi f$, $A$ is the area of the WE and $D_O$ and $D_R$ are respectively the diffusion coefficients of O and R. At high frequencies, $Z_w \to 0$, meaning that the diffusional process is too slow to influence the current. The characteristic $\omega^{-1/2}$ roll-off of $Z_w$ prevents the use of simple circuital elements; however, $RC$ approximations of $Z_w$ are possible once a range of frequencies of interest is specified [102].

A typical approximated Nyquist plot (Imaginary vs. Real part) deriving from the Randles model is depicted in Figure 5b. It is worth noting that the series configuration of $R_{ct}$ and $Z_w$ reflects the current limitations of the faradaic process. When $R_{ct}$ dominates, the system is kinetically limited, while, when $Z_w$ dominates, the system is limited by the mass transfer phenomenon. The operating condition of the cell, i.e., the value of $V_{cell}$, determines which phenomenon is dominating.

To facilitate the discussion on the CMOS interfaces, we analyse here the spectral properties of the electro-chemical cell by introducing the following time constants:

$$\tau_k \doteq R_{ct}C_{dl}; \quad \tau_d \doteq (\sigma C_{dl})^2. \tag{8}$$

$\tau_k$ characterises the exponential relaxation when mass transport is negligible, while $\tau_d$ takes into account the diffusional process. Table 3 reports the expression for the WE impedance ($Z$) as a function of $\tau_k$ and $\tau_d$, while Figure 6 shows the Bode and Nyquist plots for different values of $\tau_k/\tau_d$.

**Table 3.** Expressions of the real and imaginary parts of the electrochemical impedance of the Randles circuit in Figure 5.

| | | Expressions [91] | Expressions after Equation (8) |
|---|---|---|---|
| Re$\{Z(\omega)\}$ | $R_\Omega +$ | $\dfrac{R_{ct} + \sigma/\sqrt{\omega}}{(C_{dl}\sigma\sqrt{\omega} + 1)^2 + \omega^2 C_{dl}^2(R_{ct} + \sigma/\sqrt{\omega})^2}$ | $R_\Omega + \dfrac{1}{C_{dl}} \cdot \dfrac{\tau_k + \sqrt{\tau_d/\omega}}{(\sqrt{\tau_d\omega} + 1)^2 + \omega^2(\tau_k + \sqrt{\tau_d/\omega})^2}$ |
| Im$\{Z(\omega)\}$ | | $\dfrac{\omega C_{dl}(R_{ct} + \sigma/\sqrt{\omega})^2 + \sigma/\sqrt{\omega}(C_{dl}\sigma\sqrt{\omega} + 1)}{(C_{dl}\sigma\sqrt{\omega} + 1)^2 + \omega^2 C_{dl}^2(R_{ct} + \sigma/\sqrt{\omega})^2}$ | $\dfrac{1}{C_{dl}} \cdot \dfrac{\omega(\tau_k + \sqrt{\tau_d/\omega})^2 + \sqrt{\tau_d/\omega}(\sqrt{\tau_d\omega} + 1)}{(\sqrt{\tau_d\omega} + 1)^2 + \omega^2(\tau_k + \sqrt{\tau_d/\omega})^2}$ |

At this point, it is interesting to determine the frequency regions dominated by the two phenomena. It is evident in the Bode plot that the $\tau_k/\tau_d$ ratio strongly influences the frontier between the $Z_w$-dominated and the $R_{ct}C_{dl}$-dominated regions. For $\tau_k$ in the order of $\tau_d$, $|Z|$ presents a double slope characteristic: at low frequencies, $\omega \leq \tau_d^{-1}$, $|Z|$ presents a $-10$ dB/dec roll-off, indicating the predominance of $Z_w$. For $\omega > \tau_d^{-1}$, the slope starts to follow a $-20$ dB/dec slope.

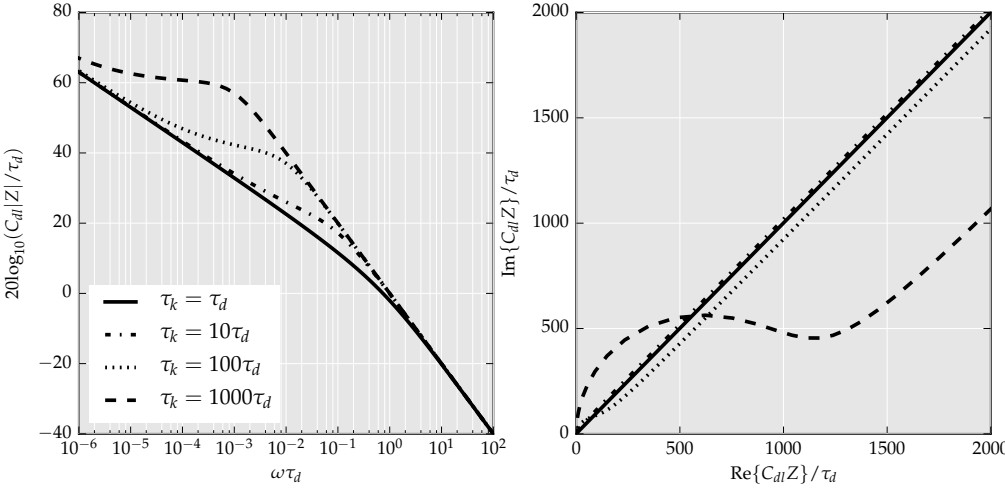

**Figure 6.** Bode (left) and Nyquist (right) plots for WE impedance $Z$. $R_\Omega = 0$ in all cases.

If the electrode kinetic is sluggish, i.e., $\tau_k \gg \tau_d$, a more complex behaviour is manifested, especially at low frequency. In this case, the $\tau_k^{-1}$ determines the pole location as in a standard *RC* circuit. At lower frequencies, the intersection with $Z_w$ is still present, but it may occur far from the bandwidth of interest.

## 4. Advanced CMOS Interfaces for Electrochemical Sensors

### 4.1. Potentiometric Interfaces

Potentiometric sensing, based on the two-terminal measurement setup of Figure 2, relies on (1) to assess the concentration of a target analyte. The readout scheme of Figure 7, employed in the analog front-end (AFE) of the Melexis MLX90129 transponder (Melexis N.V., Ypres, Belgium), features two programmable gain amplifier (PGAs) to accommodate the weak potentiometric signal for the input fullscale of the ADC, after the removal of the constant baseline through a dedicated digital-to-analog converter (DAC). The system depicted in Figure 7, while providing a versatile architecture for a number of application cases, entails a certain degree of complexity. A simpler readout chain may be considered by removing the DAC and adopting a single amplification stage. This comes at the cost of increased resolution requirements for the ADC since now its fullscale is partially occupied by the constant term in Equation (1).

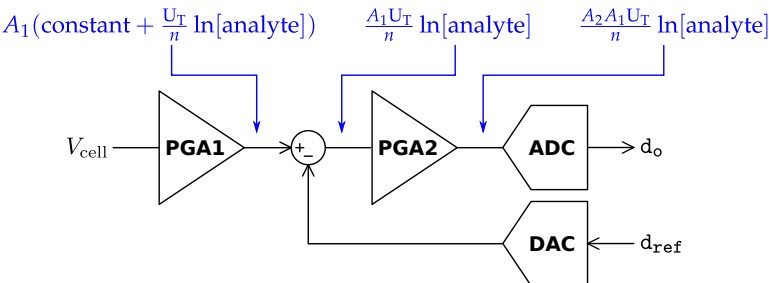

**Figure 7.** Generic potentiometric readout chain.

A detailed look at Table 1 reveals that, in some cases, specifications of the readout interface may require challenging resolutions in terms of input voltage levels. For instance, in the case of pH assessment in blood, a maximum variation <5 mV of the transduced signal is expected from Equation (1) considering the normal ranges reported in Table 1. Such weak signals are distributed in the sub-hertz frequency range where circuits' offset, offset thermal drift and flicker noise need to be cancelled in order to achieve and accurate and precise measurement.

Well-known circuital techniques for offset and low-frequency noise reduction in CMOS circuits are autozero (AZ), correlated double sampling (CDS) and chopper stabilization (CHS) [103]. Among them, CHS is immune to wideband noise fold-over into DC since it does not employ sampling. A part from intrinsic circuits' noise, environmental electrical interferences and spurious signals spread along circuits' bandwidth may be aliased in the base-band if they find a coupling path through the sampler. This could be easily the case of IoW devices, where large area sensors are coupled to their readout interfaces through printed conductive tracks onto a flexible substrate. Even in the ideal case of not having interferers, CHS is still to be preferred because it leaves a background noise equal to only one replica of the wideband thermal noise, while CDS and AZ, constrained by the bandwidth over sampling-rate ratio, would leave a certain multiple, usually $\geq 3$, of the same background level.

Figure 8a shows a classical readout chain based on chopper stabilized instrumentation amplifier (In-amp). The input fully-differential modulator, usually implemented with four cross-coupled switches controlled by a two-phase, 50%-duty cycle digital clock, provides the up-conversion of the potentiometric signal around the chopper frequency $f_{ch}$ and its odd multiples. The effect is equivalent to multiplying the signal by a dimensionless unitary square waveform, indicated with $m(t)$ in the figure. To fully benefit from CHS, $f_{ch}$ should be high enough to move the modulated signal in a region where the flicker noise is not any more dominant over the wideband thermal noise.

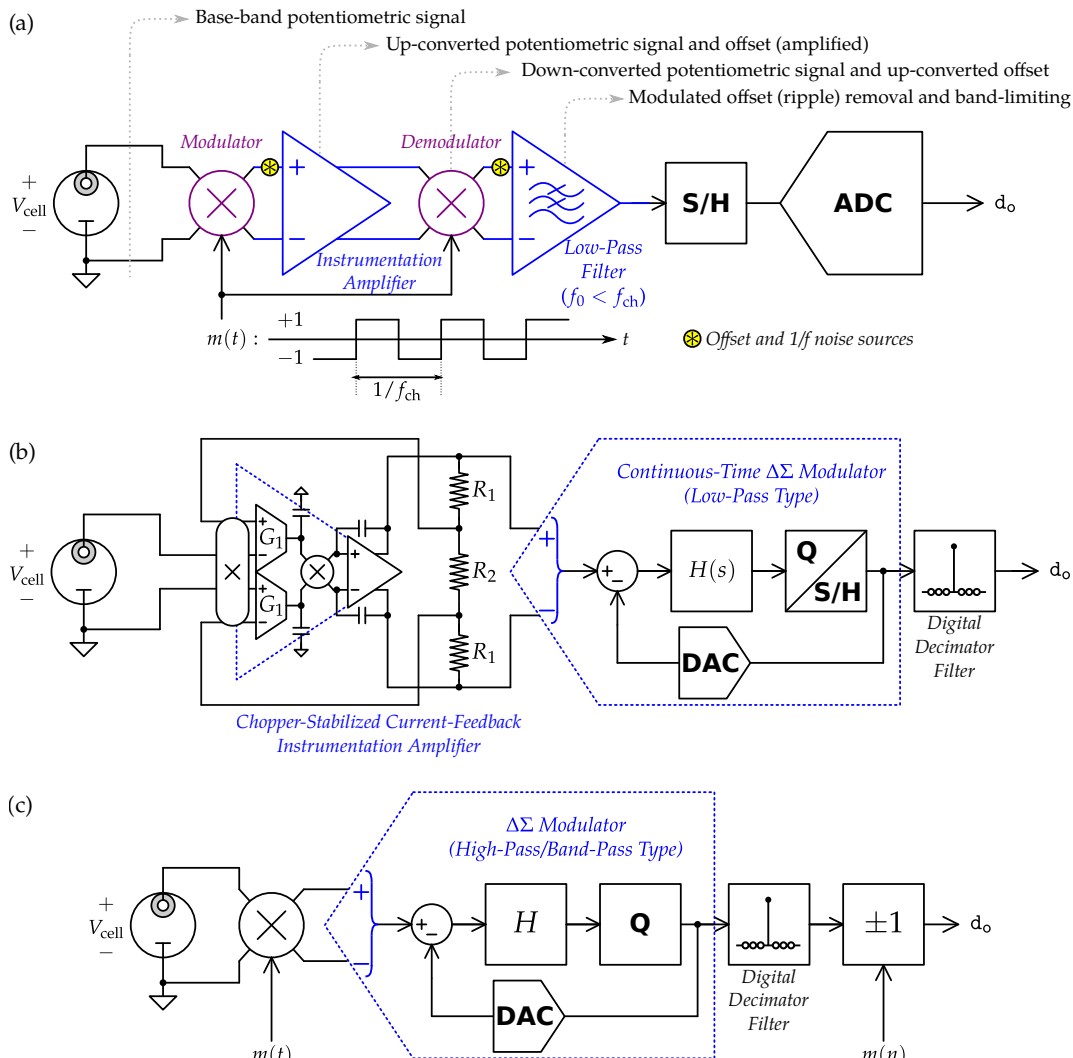

**Figure 8.** Readout chain variants for potentiometric signals: (**a**) classic chopper-stabilized redout chain; (**b**) improved by means of a modified current–feedback instrumentation amplifier; (**c**) based on early digitalization and system-level chopper stabilization.

However, several trade-offs influence the choice of $f_{ch}$: high chopper frequencies offer better flicker noise rejection and less area requirements for the In-amp and the subsequent low-pass filter [104,105], whereas low chopper frequencies reduce the residual offset and power requirements of the whole interface. In practice, chopper frequencies of few tens of kHz are used.

The In-amp processes, i.e., amplifies, its own offset and the upconverted potentiometric signal. The cascaded demodulator reverts the modulation by bringing back the potentiometric signal into the baseband and upconverting the amplifier offset. At this point, the offset which appears as a zero mean value square wave signal of $\pm A \cdot V_{OS}$ amplitude, where $A$ is the In-amp precise gain and $V_{OS}$ its input referred offset voltage.

A low-pass filter is needed in this case to interface the sample and hold circuit of the ADC for band-limiting and modulated component offset removal. At this stage, low-frequency noise and offset are added again to the potentiometric signal by the filter and the ADC, but their effect is less important since now the signal has been amplified. Nevertheless, the practical limitations apply since the the $A$ is limited by the In-amp output fullscale and the filter input fullscale, which must not be exceeded by the sum of the amplified potentiometric signal and amplified modulated offset. This scenario is aggravated in the case of processing multiple channels, in a frequency multiplexing fashion [106].

A more advanced architecture, shown in Figure 8b, avoids the use of a low-pass filter by combining advanced chopper-stabilized In-amps, capable to auto-suppress the offset ripple, and the eventual use of continuous-time $\Delta\Sigma$-modulator which has intrinsic anti-aliasing characteristics [107]. Regarding advanced In-amps, a variety of solutions has been proposed in the last several years comprising selective band-pass In-amps [108] and current–feedback amplifier topologies featuring: ripple rejection feedback loops [109–112], use of switched-capacitors (SC) filters [113,114], ping-pong autozeroing [115], embedded low-pass filter [116,117] or digital calibration techniques [118,119]. Among all these techniques, the digitally-assisted solutions [118,119] proved to be the more efficient in terms of area and power. The ripple rejection technique proposed in [112] also promises good area efficiency, but silicon implementation has not yet been demonstrated.

A very important characteristic of CHS In-amps is their input impedance, which should be ideally infinite to avoid sensor loading. This is particularly true for ISEs, which are based on the Nernstian equilibrium of Equation (1) implying a zero net current flow at electrodes. Unfortunately, due to the combined effect of modulator activity and input parasitic capacitance of the In-amp, a net non-zero current is drawn from the sensor. Figure 9 helps understand this phenomenon. At each $m(t)$ transition, the input differential capacitance $C_p$ undergoes a total voltage variation equal to $2V_{\text{cell}}$, accounting then for a charge $Q_p = 2C_pV_{\text{cell}}$ drawn from the sensor. When the opposite $m(t)$ transition occurs, approximately the same charge drawn from the sensor, an equivalent input resistance is estimated by

$$R_{in} = \frac{V_{\text{cell}}}{2Q_pf_{\text{ch}}} = \frac{1}{4f_{\text{ch}}C_p}. \tag{9}$$

Considering $f_{\text{ch}} = 10\,\text{kHz}$ and an optimistic $C_p = 500\,\text{fF}$, Equation (9) estimates $R_{in} = 50\,\text{M}\Omega$, which would already imply a considerable current leakage flowing through the WE with the adverse effect of degrading the measurement precision.

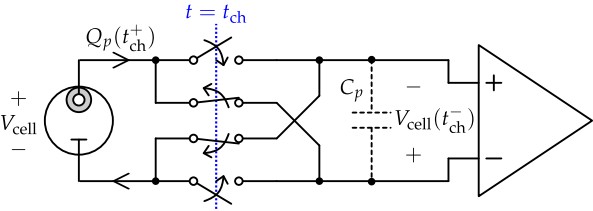

**Figure 9.** Potentiometric cell loading during transition of the input chopper modulator.

The chopped stabilized current–feedback amplifier proposed in [117] is particularly interesting because the swapping strategy of the input feedback ports of the In-amp demonstrated an input resistance boosted up to values exceeding $1\,\text{G}\Omega$ when the amplifier is operated at $f_{\text{ch}} = 20\,\text{kHz}$.

System level chopping, sketched in Figure 8c, is a quite mature concept [120,121]. It postulates the application of CHS around a generic ADC: the modulator operates on the input analog signal while the demodulator operates on the output converted digital signal. While input modulation can be achieved through the cross coupled switches, the demodulation operates directly in the digital representation of the signal by changing the sign of the the data. System level chopping is very attractive since it enables early digitalization of the potentiometric signals greatly simplifying the readout chain and consequently enabling area and power reduction. A high-resolution readout can be achieved using a $\Delta\Sigma$ modulator, which needs to now deal with the up-converted potentiometric signal around at $f_{\text{ch}}$ and its odd multiples. Consequently, a band-pass/high-pass modulator type is needed instead of the traditional low-pass type. A general rule to derive pass-band modulators from their low-pass model, is to substitute the integrator blocks with their resonator counter parts.

When an SC solution is considered, the resonator blocks are efficiently realized through the pseudo-2-path SC resonators [107] (p. 156). The same block applied to a CHS $\Delta\Sigma$ converter has been recently analyzed to highlight its functional characteristics for various $f_{\text{ch}}/f_{\text{sampling}}$ ratios [122].

An interesting silicon implementation of the same idea is provided in [123] consuming only 144 µW and reducing the input offset to 403 µV. It is worth noting that SC implementations of the concept in Figure 8c, where no anti-aliasing filter is present, may introduce significant degradation of the converter resolution due to noise fold-over.

Alternatively to ΔΣ-modulators, Successive Approximation Register (SAR) ADCs can be employed for digitalization. Ref. [124] presents a potentiometric AFE based on an SAR ADC constrained to a power budget for the sensor AFE of <1.2 pW and capable of 8.3 ENOB of resolution. Such a remarkable ultra-low power feature is achieved thanks to a reference-free charge-sharing SAR architecture which avoids the use of power hungry voltage reference buffers [125]. In addition, since an extremely slow sample rate is used (10 S/s), a combination of two different and clock generation strategies is used. The sampling is controlled by an ultra-low-power capacitve discharging oscillator running at 10 Hz [126], while the conversion-step clock is generated by an asyncronous unit running at 1 MHz for the 10 controlling slices of the conversion process [127]. This strategy allows for greatly relaxing leakage problems of the charge sharing architecture when operated at extremely low switching frequency.

Despite the great amount of techniques and circuital possibilities already known, research efforts are still needed to find optimal potentiometric interface solutions for the IoW.

### 4.2. Amperometric Interfaces

A potentiostatic interface performs three functions: (i) it sets the cell operating voltage $V_{\text{cell}}$; (ii) it reads out the redox current at WE, $I_W$; and (iii) it provides analog-to-digital (A/D) conversion. Figure 10 presents general topologies for potentiostats that perform the amperometric readout relying on current to voltage ($I_W$-to-$V_A$) conversion through a resistor $R$ prior its A/D conversion [128–131]. The most common configuration is shown in Figure 10a. Here, $A_1$ and $A_2$ set $V_{\text{RE}}$ and $V_{\text{WE}}$, respectively, through local feedback loops. While $A_1$ feedback is built around the sensor to set the RE potential, $A_2$-$R$ realize the well-known trans-impedance amplifier (TIA) configuration while setting at the same time the WE potential. If $A_1$ and $A_2$ are ideal operational amplifiers, OpAmps, straightforward circuit analysis reveals

$$V_{\text{cell}} \doteq V_{RE} - V_{WE} \approx V'_{RE} - V'_{WE}; \quad V_A = RI_W, \tag{10}$$

where the high-gain operational amplifiers ($A_1 \gg 1$ and $A_2 \gg 1$) ensure that $V_{RE} \approx V'_{RE}$ and $V_{WE} \approx V'_{WE}$. Equation (10) shows that $V_{\text{cell}}$ can be set by independently varying $V'_{RE}$ or $V'_{WE}$. The same equations apply to the circuit in Figure 10b. However, in this configuration, the sensor in included in both $A_1$ and $A_2$ loops. More importantly, the feedback current of $A_2$ in Figure 10b flows through both $R$ and the sensor. This way, the sensor is included in the readout loop, giving rise to very compact topologies that will be commented on at the end of this Section.

The potentiostats of Figure 10a–c provide a very wide swing of $V_{\text{cell}}$, ideally doubling the effective supply voltage, provided that the DAC outputs, $V'_{RE}$ and $V'_{WE}$, can swing across the full span of the voltage supply rails. It is also clear that the common-mode input range of the amplifiers in schemes (a) and (b) must also be rail-to-rail since one of the inputs is connected to either $V'_{RE}$ or $V'_{WE}$. Nevertheless, voltage limitations on the $V_{\text{cell}}$ swing are evident once the output of $A_2$ ($V_{out,A_2}$) is considered. Without loss of generality, we assume that a single supply voltage (rails gnd, $V_{\text{DD}}$) is available. For both schemes (a) and (b),

$$V_{out,A_2} = V_{WE} - RI_W. \tag{11}$$

Let us first consider the case of $V_{\text{cell}} > 0$ and $I_W > 0$: in this situation, $V_{out,A_2}$ tends to lower voltage values, in order to avoid saturation of $A_2$,

$$V_{out,A_2} > 0 : V_{WE} - RI_W = V_{RE} - V_{\text{cell}} - RI_W > 0 \Rightarrow V_{RE} > V_{\text{cell}} + \underbrace{RI_W}_{V_A}. \tag{12}$$

By conveniently setting $V_{RE} \to V_{DD}$, the maximum room for voltage allocation between $V_{cell}$ and $V_A$ can be obtained. Conversely, for the case of $V_{cell} < 0$ and $I_W < 0$, $V_{out,A_2}$ tends to saturate towards $V_{DD}$:

$$V_{out,A_2} < V_{DD} : V_{WE} + R|I_W| = V_{RE} + |V_{cell}| + R|I_W| < V_{DD} \Rightarrow V_{DD} - V_{RE} > |V_{cell}| + \underbrace{|RI_W|}_{|V_A|}. \quad (13)$$

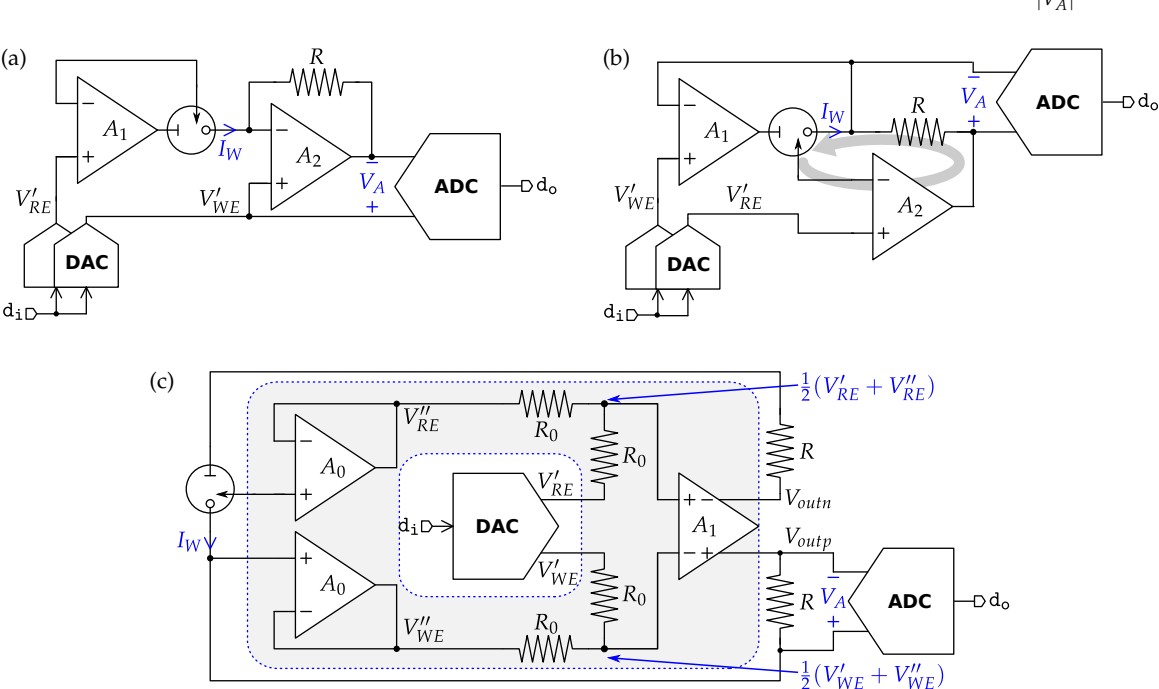

**Figure 10.** Potentiostatic interfaces based on continuous time *I*-to-*V* conversion (**a**) conventional solution; (**b**) sensor-in-the-loop solution; (**c**) fully-differential solution.

Thus, voltage range is maximized by setting $V_{RE} \to 0$. Equations (12) and (13) clearly show the trade-offs to be considered, between the potentiostatic range ($V_{cell}$) and the trans-resistive gain, $R$, that can be achieved. This trade-off is particularly severe when a low-voltage design is targeted.

Fully differential topologies such as those in Figures 10c have also been proposed [131–133]. The solution in [133] replaces all the gray area in Figures 10c with a fully-differential difference amplifier (FDDA). Analytically, considering the Randles model of Figure 5 and considering that no redox process is present at CE,

$$V_{CE} = V_{outn} - RI_W \approx V_{RE} + R_\Omega I_W; \quad V_{WE} = V_{outp} + RI_W. \quad (14)$$

Now, expressing each differential output voltages $V_{outn}$, $V_{outp}$, it will be easily found that

$$V_{cell} = \frac{A_1}{1 + A_1}(V'_{RE} - V'_{WE}) - \frac{(2R + R_\Omega)I_W}{1 + A_1} \underset{(A_1 \gg 1; R \gg R_\Omega)}{\approx} V'_{RE} - V'_{WE} - \frac{2V_A}{A_1}. \quad (15)$$

Equation (15) reveals that both $R_\Omega$ and $R$ play a detrimental role on the precision of potentiostat operation, and can only be compensated by large values of $A_1$. While $R_\Omega \ll R$, and it can thus be neglected in many practical cases, the issue is still present due to the fact that $R$ is usually as large to accommodate $V_A$ to the input full scale of the ADC. Therefore, the error on $V_{cell}$ can be approximated to $2V_A/A_1$. Moreover, scheme (c) is still subject to the trade-offs of Equations (12) and (13).

The same equations can be reinterpreted to find the specifications for the input common mode swing of the ADCs. All the schemes shown in Figure 10 imply demanding requirements for input common mode range as it needs to span from 0 to $V_{DD}$ to provide the maximum voltage room for

$V_{\text{out},A_2}$. The use of an instrumentation amplifier, as interstage between the *I*-to-*V* stage and the ADC, may provide a suitable common mode matching and further gain. On one hand, distributing the gain among various stages may result in being beneficial for medium/high bandwidth application [134], but this could rarely be the case for IoW interfaces. On the other hand, *R* also defines the noise floor asymptote of the amperometric readout, which finally determines the amperometric limit of detection (LOD) of the potentiostat:

$$S_{i_n} = \frac{4KT}{R}; \quad \text{LOD}_{\text{ideal}} = 3\sqrt{S_{i_n} \times \text{bandwidth}}. \tag{16}$$

In practice, extremely large values of *R* ($>10$ MΩ) are avoided since the integration of such high resistances is too expensive in terms of silicon area. The use of off-chip resistors is feasible, but it comes at the cost of extra pads. By looking at Equation (5), it can be seen that the amperometric sensitivity can be incremented by enlarging the sensing area of the WE, which is already an off-chip component of the envisioned IoW system, thus avoiding the use of extremely large *R* and relaxing the LOD requirements.

Amplifiers' noise is also present and their effect in the amperometric readout has been studied in [135] for simplified Randles circuit ($Z_w = 0$). From analysis in Section 3.3, this is the case of a sluggish process, i.e., when the electrode kinetics are dominated by the charge transfer mechanism. More generally, a beneficial interplay between the sensor impedance and the $1/f$ noise components exists at low frequencies and will be addressed here. In Figure 11, referred-to-input noise of $A_1$, $A_2$ has been introduced. Small-signal equations can be written:

$$\begin{cases} i_n = (v_{CE} - v_{WE})/Z \\ v_{CE} = R_\Omega i_n + v_{RE} \end{cases} \longrightarrow \quad i_n = \frac{v_{CE} - v_{WE}}{Z - R_\Omega} : \quad S_{i_n} = \frac{S_{v_1} + S_{v_2}}{|Z - R_\Omega|^2}. \tag{17}$$

**Figure 11.** Noise sources in a transimpedance-amplifier based amperometric readout. Configuration (**a**) of Figure 10 is considered here.

Since $v_{n1}$, $v_{n2}$ are originated from different amplifiers, they should be considered as uncorrelated noise sources. At very low frequencies, $f < f_c$, where $Z \approx Z_w$, we can use the flicker noise expressions for $S_v$:

$$S_v|_{f<f_c} = k_A \frac{K_F}{WL} \frac{1}{|f|} \longrightarrow \quad S_{i_n}|_{f<f_c} = \frac{K_F}{4\sigma^2} \left[ \frac{k_{A_1}}{(WL)_1} + \frac{k_{A_2}}{(WL)_2} \right], \tag{18}$$

$K_F$ being the flicker coefficient of the MOS transistors for the given technology; $W$, $L$ the geometric characteristics of the amplifiers' input devices and $k_{A_1}$, $k_{A_2}$ the respective excess noise factors of $A_1$ and $A_2$ taking into account the circuital implementation. Equation (18) shows clearly that the Warburg impedance provides built-in flicker noise stabilization of the electronic interface. It is important to notice that $\sigma$ depends on the analyte concentration (both in its reduced and oxidation state), so the total noise power is expected to be signal dependent.

Another noise contribution must be accounted for when considering the readout point after the TIA stage. Any fluctuation caused by $v_{n2}$ propagates to the output of the TIA regardless of $Z$, so Equation (17) underestimates the total noise contribution. A more complete expression can be found by direct inspection of the circuit of Figure 11:

$$
\begin{cases}
v_{A,\text{noise}} = \left(1 + \dfrac{R}{Z - R_\Omega}\right) v_{n2} - \dfrac{R v_{n1}}{Z - R_\Omega} \\[2ex]
i_{n,\text{rti}} = \dfrac{v_{A,\text{noise}}}{R}
\end{cases}
\longrightarrow
S_{i_{n,\text{rti}}} = \frac{S_{v_1} + S_{v_2}}{|Z - R_\Omega|^2} + \left(1 + \frac{2R}{\text{Re}\{Z - R_\Omega\}}\right)\frac{S_{v_2}}{R^2}. \quad (19)
$$

A multifaceted spectral scenario must be considered for the calculation of the LOD once the amplifiers' noise contribution is taken into account:

$$
\text{LOD} = 3 \times \left[ \int_{T_{obs}^{-1}}^{\text{bandwidth}} S_{i_{n,\text{rti}}}(f)\, df \right]^{\frac{1}{2}} + \text{LOD}_{\text{ideal}}. \quad (20)
$$

Here, $T_{obs}$ is the observation time of the measurement. Practical limits apply to the validity of Equation (18) for very large $T_{obs}$, and the diffusion layer is so extended that other phenomena detrimentally influence the measurement. Among such phenomena, the most relevant are convection and sample evaporation. Due to the high flicker noise levels present in CMOS circuits, high accuracies require the use of dynamic techniques for $1/f$ noise reduction, such as CHS, AZ or CDS.

Another well known continuous time amperometric readout technique is based on current mirror amplifiers [129,130,134,136–139], such as those shown in Figure 12. A closer look at the configurations in Figure 10 reveals that the potentiostatic control amplifier $A_1$ provides the current to the sensor, so it can be copied and eventually amplified by simple current mirror configurations.

In Figure 12a, the mirror is embedded in the amplifier, and current is precisely copied with the help of an auxiliary amplifier $A_2$ and a cascode transistor ($M_C$) [129]. The sensor current is amplified thanks to the multiplier factor $m$ of the current mirror. It is worth noting here that the sensor readout current biases $M_1$, which also provides an additional gain stage in the amplification loop. Thus, when the current is extremely low, as in case of calibration with a blank sample, the control loop may be easily pushed to instability. A minimum bias current $I_0$ can be added at the input of the mirror to circumvent this problem.

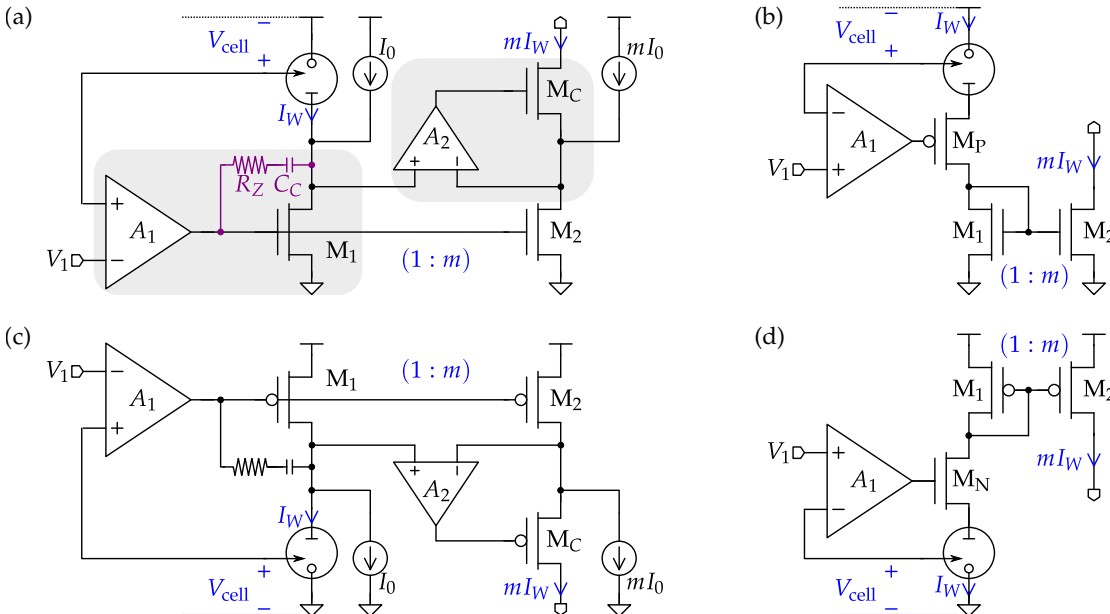

**Figure 12.** Potentiostats based on current mirror amplifiers. (**a**) n-type mirror embedded in the amplifier; (**b**) n-type mirror with current buffer; (**c**) p-type mirror embedded in the amplifier; (**d**) p-type mirror with current buffer.

Stability can be further enhanced by introducing an LHP zero in a feed-forward signal path around $M_1$ [136]. Schemes (a) and (c) shows a simple zero-nulling through a resistor $R_Z$; however, other optimum compensation schemes may be employed depending on the actual circuital implementation of $A_1$ [140].

Configuration in Figure 12b employs a p-type device $M_P$ acting as a common drain stage rather than a common source stage, thus providing a current buffer between the control loop and the current mirror to improve stability. The simpler topology of Figure 12b comes at the cost of reduced headroom for $V_{cell}$ due to the $V_{GS}$ drop of $M_P$. While both topologies are attractive for high-bandwidth applications and for the low-count of devices, noise performance can be of concern since $1/f$ components of $M_1$-$M_2$ are not filtered through the impedance of the sensor and they sum up directly to $I_W$. On the other hand, their potentiostatic capabilities are also limited by the fact that the WE is fixed to the positive supply voltage, so only negative $V_{cell}$ can be applied. The complementary topologies of Figure 12c,d, i.e., interchanging n-type and p-type devices, are also possible, which would allow the application of positive values of $V_{cell}$ only.

High-sensitive readout can be achieved by replacing the resistive feedback element with a capacitor, following the SC paradigm. Intuitively, $R$ in Equation (19) is now substituted by $1/(j\omega C)$, so $S_{v2}$ noise contribution is greatly attenuated in the low frequency range. In such case, we can express $V_A$ between the integration frame delimited by $T_{i-1}$ and $T_i$ as

$$V_A(T_{i-1}, T_i) = \frac{1}{C} \int_{T_{i-1}}^{T_i} I_W(t)\, dt = \frac{T_i - T_{i-1}}{C} \cdot \overline{I_W}, \tag{21}$$

which reveals that the readout value $V_A$ holds the averaged $I_W$ current within the integration frame.

The use of a capacitive element allows the I-to-V interface to reduce area occupation, and to benefit from better linearity and less PVT variations with respect to resistive elements. However, a sustained current would soon drive to saturation the integrator so some sort of reset strategy is needed [141–147].

A classic SC reset mechanism that also performs CDS is shown in Figure 13 [142,143,147]. The scheme in (a) features a PGA built around $A_3$, whose (inverting) gain is set by $C_1/C_2$. Here, during the reset phase $\varphi_1$, $v_{n2}$ and $v_{n3}$ are sampled respectively onto $C_1$ and $C_2$.

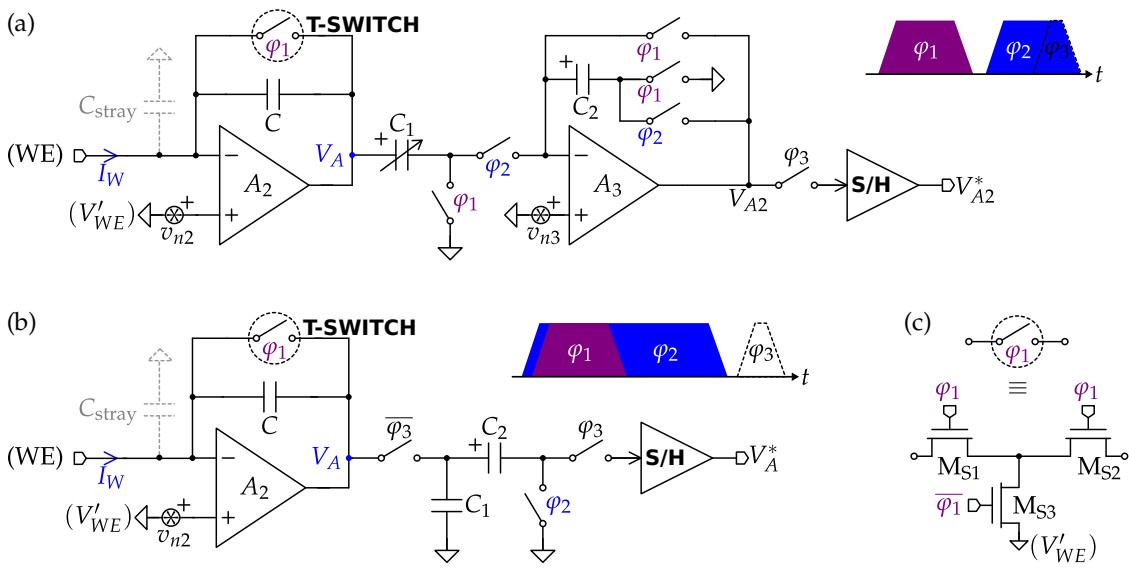

**Figure 13.** Switched-capacitors amperometric readout schemes including flicker noise cancellation by correlated-double sampling. (**a**) conventional scheme; (**b**) without the programmable-gain amplifier; (**c**) T-switch schematic.

In the integration phase $\varphi_2$, the signal follows the integration and the amplification stages together with the current values $v_{n2}$ and $v_{n3}$, which now contributes with the opposite sign. Noise sample differentiation of the CDS techniques is thus readily implemented. At the end of $\phi_2$, the sample and hold (S/H) phase $\phi_3$ delivers the processed output $V_{A2}^*$ to the following A/D stage. Scheme (b) follows the same philosophy as (a), without the PGA stage [135]. A different arrangement of the phases allows for signal integration and noise differentiation with only one active element.

While SC readout chains offer the switching frequency as a valuable degree of freedom to the designer, especially to avoid integration of large $R$ on chip, the achievable resolution is affected by several issues, which are specific to SC circuits, such as: noise folding due to the sampling process at $C_1$ and $C_2$ ($kT/C$ noise), amplifier noise, charge injection, clock feed-through and reset switch leakage at the integrator stage.

Noise folding can be cut down by setting relatively large values of $C_1$ and $C_2$. As detailed in [148] and references therein, amplifier noise can be embodied into $kT/C$ noise as a form of excess noise that can be limited by a proper design. In practice, values of $C$ in the order of $\approx 100$ fF are sufficient to make $kT/C$ noise contributions smaller than ADC resolution. Charge injection and clock feed-through can be reduced through fully-differential architectures, dummy switches [103] and advanced layout techniques [149]. Leakage of the reset switch can be minimized by using the t-switch configuration shown in Figure 13c [150]. In this case, leakage currents of $M_{S1}$ are nulled by equalizing the source/drain voltage through the virtual ground on the left side and the shunt connection through $M_{S3}$ on the right side.

So far, the A/D conversion step in the signal processing flow has been considered as a separate function carried out by a dedicated block, namely the Analog-to-Digital Converter (ADC). This classic approach is followed in [146], where an SAR ADC has been coupled to an SC-integrator based readout chain. SAR ADCs represent a popular solution in the medium-low resolution range thanks to their excellent power efficiency. On the other hand, ADC based on $\Delta\Sigma$ modulators is also an attractive choice for high resolution low-bandwidth applications like the IoW. While $\Delta\Sigma$ ADCs needs a digital decimation filter for signal reconstruction from the $\Delta\Sigma$ bit stream produced by the modulator, they employ the oversampling ratio (OSR) to relax both thermal noise and matching constraints.

At the same time, the need for early digitalization inspired a number of compact readout architectures providing fully or semi digital outputs. Figure 14 illustrates the main difference in potentiostatic functioning principles between the classic and the compact approach. The compact approach skips the $I$-to-$V$ conversion step in favour of direct $I_W$ digitalization. In some cases that will be discussed next, the same digital output is fed back to the potentiostat in order to set the $V_{\text{cell}}$. In the remainder of this section, potentiostats will be classified as:

1. Digital potentiostat by the means of Dual-Slope A/D conversion [141,150];
2. I-to-f potentiostats: the readout current is converted to a square-wave signal whose frequency is proportional to the current amplitude [136–139,151];
3. I-to-PDM (pulse density modulation) potentiostats: the readout current modulates the density of a pulse repetition in time [128,152–154];
4. I-to-$\Delta\Sigma$M potentiostats: the readout current is converted into a stream of $\Delta\Sigma$ modulated signals [150,155–161].

Smart potentiostats falling into the first category employ a capacitive TIA (CTIA) stage whose input is fed alternatively by the $I_W$ current and a reference discharge current $I_{\text{ref}}$. Figure 15 shows the simple circuital arrangement and the timing of the dual-slope configuration. During the integrating phase, $I_W$ charges $C$ either to cause a positive or negative ramp of $V_C$. The total integrated charge will be $T_{\text{int}} I_W$. At the end of this phase, the comparator decides the polarity of $I_W$ in order to determine the $I_{\text{ref}}$ to be of source or sink type during the controlled discharge phase. A digital counter, embedded in the digital control block, starts to run until the comparator reverts to its initial state; at this point, the $I_{\text{ref}}$ stops integrating onto $C$, thus the charge variation during this phase will be $T_{\text{dis}} I_{\text{ref}}$ in magnitude,

and its sign will be opposite with respect to the charge accumulated during integration. Since the comparator triggers on the same voltage level, the net charge change over $C$ in the current conversion cycle is zero ($T_{\text{int}} I_W = T_{\text{dis}} I_{\text{ref}}$), and the counter output will be represented by the digital value of:

$$\mathtt{d_{out}} = f_{\text{ck}} T_{\text{dis}} = f_{\text{ck}} T_{\text{int}} \frac{I_W}{I_{\text{ref}}}. \tag{22}$$

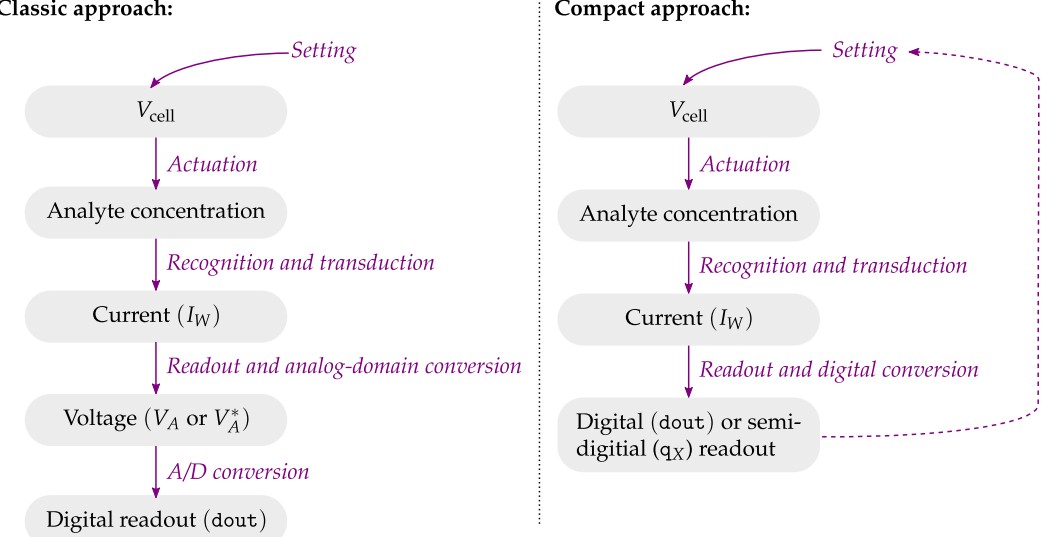

**Figure 14.** Classic and compact potentiostat approaches.

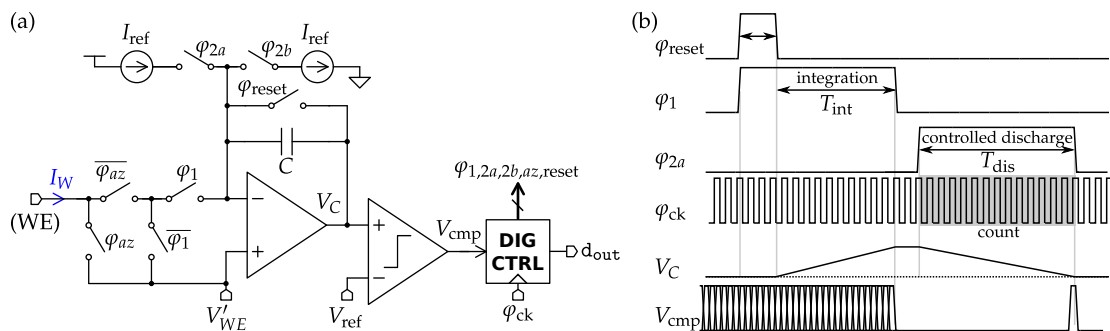

**Figure 15.** Potentiostats with embedded dual slope A/D conversion. (**a**) schematic; (**b**) time diagram.

Potentiostats based on current to frequency conversion (I-to-f) operate in a similar manner to the dual slope potentiostats, controlling the charging and the discharging of a capacitor element [128,136]. However, instead of directly converting the discharge time into a digital count, the period of a square waveform is continuously modulated which can be later converted into a digital value by time discretization.

The circuit in Figure 16a performs the I-to-f conversion through a simple relaxation oscillator constraining $V_C$ between $V_{\text{ref,H}}$ and $V_{\text{ref,L}}$. Here, $I_W$ is injected into the the I-to-f modulator through a current amplifier potentiostat such as those of Figure 12. Similarly, the circuit in Figure 16b integrates $I_W$ alternatively on two nominally identical capacitors $C_1$ and $C_2$ [138,139]. Here, the use of a single comparator triggering at a fixed voltage makes the modulated square wave insensitive to comparator offset (and to its temperature drift). Furthermore, temperature effects on the modulated signal can be cancelled out by performing a double reading, first measuring the PTAT modulated signal, and then adding the sensed current contribution:

$$f_{\text{ref}} = \frac{I_{\text{PTAT}}}{(C_1 + C_2) V_{th}}; \quad f_{\text{meas}} = \frac{I_{\text{PTAT}} + I_W}{(C_1 + C_2) V_{th}}. \tag{23}$$

The solution in Figure 16c proposes an I-to-V conversion through a starved-inverter based ring oscillator [151]. Interestingly, this interface provides a full potentiostatic control at WE through the action of a current conveyor type-II (CCII+) circuit. The voltage set at the high impedance input Y is replicated onto X providing a virtual ground; moreover, the CCII+ senses the current at the low-impedance input X and replicates it at the Z terminal. Since Z is connected to the gate of an array of transistors ($M_{R1}$–$M_{RN}$), no current is allowed to flow; consequently, the control loop imposes $I_X = 0$ and all $I_W$ is conveyed at one of the $M_R$ transistors corresponding to the starved inverter activated by the travelling wave. Only one CCII+ is needed for all the ring stages, since the regulation control of each M3 is naturally rolling along the ring in synchronism with the steering of $I_W$. This particular sequential control gives the name rolling regulated current-controlled ring oscillator to the circuit of Figure 16c. Differently from standard unregulated current-starved ring oscillators, voltage threshold to be reached by the travelling edge here is set precisely to be $V'_{WE}$. Thus, the transit time of for each ring slice is determined by:

$$T_{\text{slice}} = \frac{C V'_{WE}}{I_W},$$ (24)

and the oscillation frequency, which is a submultiple of $1/T_{\text{slice}}$, will be linearly proportional to the sensed current $I_W$. The rolling regulation allows very linear characteristics of the I-to-f conversion without the need for any digital calibration or post-compensation technique. The digital readout is performed by combining the full rounds of the travelling edge detected by the course counter, and the intermediate position of the edge inside the loop extracted by the fine encoder. This technique enables extended resolution for the same frequency range at the cost of chip area, without increasing power consumption.

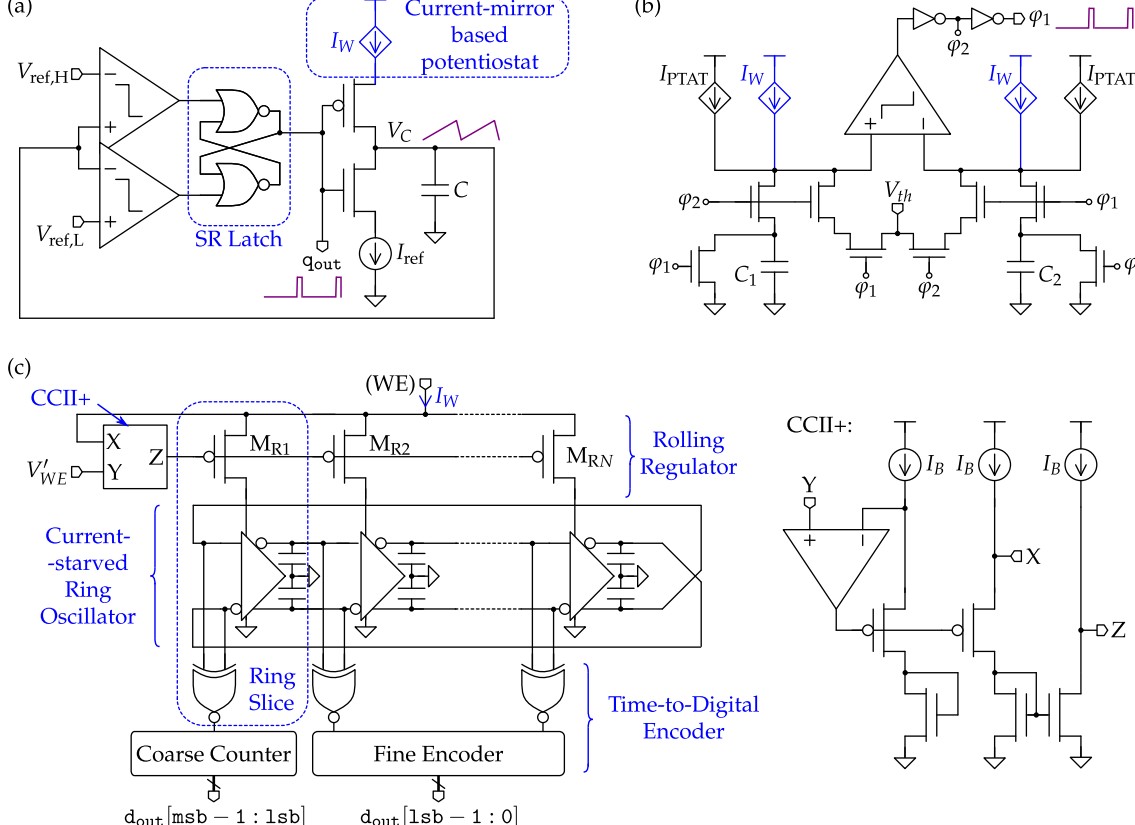

**Figure 16.** Potentiostats with embedded current to frequency conversion (I-to-f type): (**a**) based on conventional relaxation oscillator; (**b**) based on single-comparator relaxation oscillator; (**c**) based on current-starved ring oscillator.

A natural extension of of I-to-f techniques are those involving a fixed-width pulse whose repetition period is controlled by the modulating current. This way, the repetition frequency of the pulse, which is also the pulse density in time, encodes the $I_W$ information. I-to-PDM potentiostats can have a minimalistic circuital implementation as shown in Figure 17. The scheme in (a) derives from that in Figure 12b, where the current mirror is replaced by a simple integrate-and-fire (IaF) circuit composed by a shunt capacitor and a comparator [152]. Starting from a reset condition, $V_C$ increases its value at the constant rate of $I_W/C$; when $V_C$ reaches the threshold value set by $V_{DAC}$, an event is fired, whose duration $T_{pulse}$ may be controlled by the digital control circuit. At the event onset, the capacitor is reset and $T_{pulse}$ should be long enough to ensure the complete reset of the capacitor. The firing rate $f_{ev}$ is thus determined by:

$$f_{ev} = \frac{1}{\dfrac{CV_{DAC}}{I_W} + T_{pulse}} \quad \underset{(T_{pulse} \to 0)}{\approx} \quad \frac{I_W}{CV_{DAC}}. \tag{25}$$

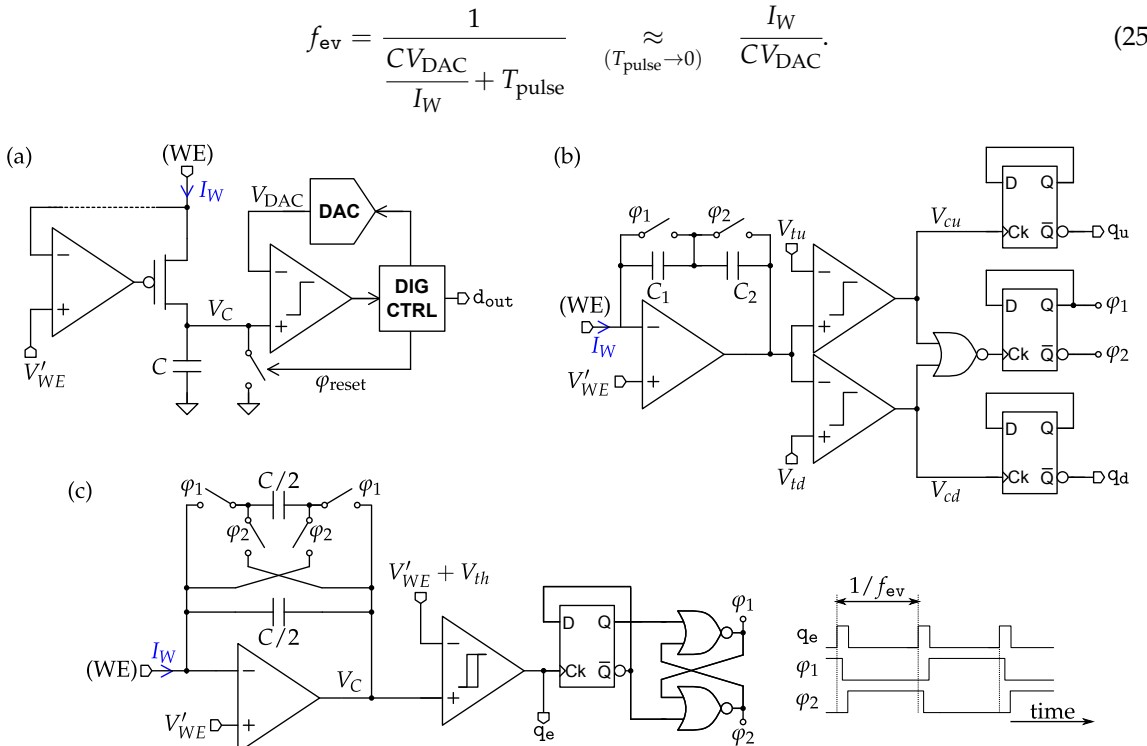

**Figure 17.** Potentiostats with embedded current to pulse density modulation (I-to-pulse-density-modulation type). (**a**) based on current buffer; (**b**) based on transimpedance amplifier with double integrating capacitor; (**c**) based transimpedance amplifier with soft-reset mechanism.

Equation (25) clearly shows that a non-zero $T_{pulse}$ implies a nonlinear $f_{ev}(I_W)$ characteristic. Thus, a very small value of $T_{pulse}$ is desirable. On the other hand, if $T_{pulse}$ is not long enough to reset $V_C$ completely, the next pulse would occur after a delay that depends not only on $I_W$ but also on the residual charge; since the residue depends on the discharging characteristic of the nonlinear MOS switch, it will not be linear with respect to its associated $V_C$ sample, finally creating a strongly nonlinear I-to-PDM conversion characteristic.

In order to maintain a sufficiently linear response for a given $T_{pulse}$, $V_{DAC}$ can be coarsely adjusted following a discrete set of $I_W$ ranges. This way, the integration time can be sufficiently larger than $T_{pulse}$ for a large range of $I_W$ values.

The circuit of Figure 17b avoids the problem related to a non-zero $T_{pulse}$ by alternating the integration and reset phase on two nominally identical capacitors $C_1$ and $C_2$ [153]. The modulator also features a double comparator to detect separately positive and negative $I_W$.

The modulator proposed in Figure 17c is also insensitive to $T_{pulse}$, without doubling the integration capacitor [154]. Instead, $C$ is split in two, and reset is performed by flipping one of the halves and thus

neutralizing the total integrated charge. More importantly, schemes (b) and (c) both perform charge neutralization without the intervention of the amplifier, greatly relaxing its speed requirements.

The last class of smart potentiostats involves those performing current to $\Delta\Sigma$-modulated bit stream. In such circuits, the signal is processed through integration and comparison to a reference voltage, as in the previous solutions. However, the comparison takes place at a constant pace; therefore, the output signal is fully digital since both its amplitude and its duration are of a discrete nature. From the circuital point of view, this avoids the use of continuous-time comparators in favour of latched-type comparators which enable important DC power reduction [162].

In the circuit of Figure 18a, the $\Delta\Sigma$ loop is built around the CTIA stage in a similar fashion to the dual-slope potentiostat of Figure 15a. However, phases' control is different here since, at the same time $I_W$ is integrated, the sinking or sourcing $I_{\mathrm{ref}}$ is also contributing to the error charge accumulated onto the integrator. At each quantization cycle, the error sign is extracted and the loop counteracts in order to cancel it. On a large time scale, the average error is nulled, thus making the averaged quantized stream $q_{\mathrm{dsm}}$ also a digital representation of the input current [155].

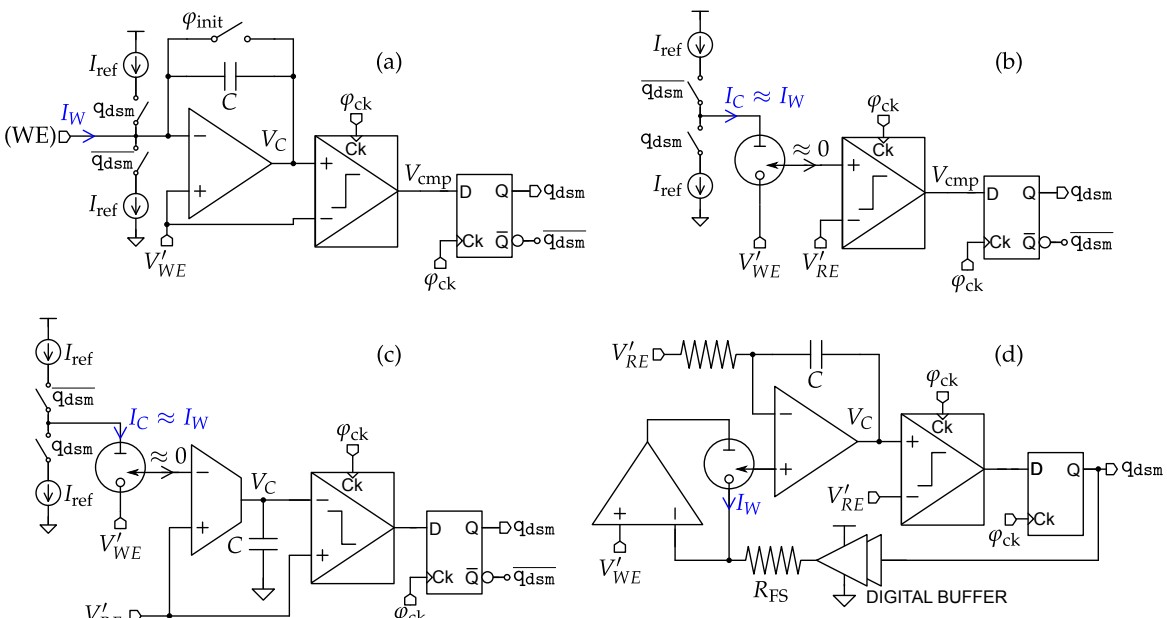

**Figure 18.** Potentiostats with embedded $\Delta\Sigma$ modulation (I-to-$\Delta\Sigma$M type). (**a**) based on transimpedance amplifier; (**b**) sensor-in-the-loop approach, first-order modulator; (**c**,**d**) sensor-in-the-loop approach, second-order modulators.

The circuit of Figure 18b also implements a $\Delta\Sigma$ loop, but this time the sensor itself, with its intrinsic low-pass quasi-integrator spectral characteristic is used as the loop shaper (see $Z$ expressions in Table 3).

The voltage error signal is extracted at RE allowing the current fed through the WE to compensate for it. This error is a small ripple around the $V'_{RE}$ set point chosen for the RE. Since the input impedance of the comparator is much smaller with respect to the intrinsic $C_{dl}$ of the cell, the alternating current provoked by the RE ripple is negligible and the condition of nearly zero current at RE is ensured. It is worth noting that scheme (b) employs the sensor-in-the-loop strategy mentioned at the beginning of this section. This way, a great compactness of the interface is obtained, which also allowed the fusion of the potentiostatic control and the amperometric readout in a single control loop.

Unfortunately, the minimalistic $\Delta\Sigma$ modulator of Figure 18b suffers from poor potentiostatic linearity [158] due to signal dependent gain of the single-bit quantizer [107]. Topologies (c) [159] and (d) [160] in the same figure solve this problem by introducing an integrating stage before the quantizer that ensures a much more precise potentiostatic control of the interface. Moreover, since now a mixed

electrochemical and pure electronic shaper is employed, the modulator order is increased providing higher signal-to-noise ratios for the same readout bandwidth. It is important here to emphasize that modulators in (b–d) implement the dashed feedback loop depicted previously in Figure 14, where the the digital readout is reused internally in the interface to set the $V_{\text{cell}}$ potentiostatic voltage.

The vast majority of the state-of-the-art works reviewed here employ uniform and continuously operating sampling. In some cases, low levels of signal for prolonged periods of times are followed by sparse burst-like activity. In another scenario, a more dense sampling is desired after the signal of interest triggers a certain threshold. In both cases, smart adaptive sampling can provide a more efficient energy management by lowering the data rate during idle periods. Typically, the very low bandwidth requirements of the many kinds of physiological signals allow for aggressive duty cycling, with large scheduled off-state periods. Thus, in energy-constrained scenarios such as the IoW, adaptive sampling rate and duty cycling will constitute the basis of the next-generation interfaces.

### 4.3. Interfaces for Sensing FETs

The idea of using a field-effect transistor (FET) as an active pH sensor was first published more than 45 years ago [163,164]. However, a major milestone in the history of the ion-selective FET (ISFET) arrived almost three decades later with the demonstration of its fabrication through unmolested standard CMOS technologies [165]. As a result, today's ISFETs benefit from the large-scale integration and low-cost manufacturing advantages of the *More Moore* and *More than Moore* technology roadmaps [166]. Recently, authors in [167] fabricated an ISFET on a flexible substrate demonstrating its employability as a wearable sweat pH sensor for healthcare.

Basically, an ISFET device is a regular MOS transistor with its floating gate isolated from the rest of the CMOS circuit but coupled to an external RE (typ. Ag/AgCl) in contact with the electrolyte solution through an insulating membrane (typ. CMOS standard passivation $Si_3N_4/SiO_2$) as the sensing area. Since no DC current is driven to the gate, the ISFET can be understood as a kind of potentiometric chemical sensor, although in most literature references its pH sensing capability is explained as a change in the threshold voltage of the corresponding floating-gate MOS device.

The equivalent electrical model of the ISFET is depicted in Figure 19. The electrochemical cell ($V_{\text{chem}}$) consists of the RE built-in potential ($V_{\text{ref}}$) together with the double-layer capacitance formed by the Gouy–Chapman layer ($C_{\text{gouy}}$) and the Helmholtz plane ($C_{\text{helm}}$) [168]. Indeed, the latter defines the pH-driven voltage drop ($V_{\text{ph}}$) at the MOSFET gate. The rest of the ISFET model of Figure 19b includes the passivation capacitance ($C_{\text{pass}}$) and the inner potential drop ($V_{\text{tc}}$) introduced by the charges trapped in the insulator layers during the MOSFET integration process.

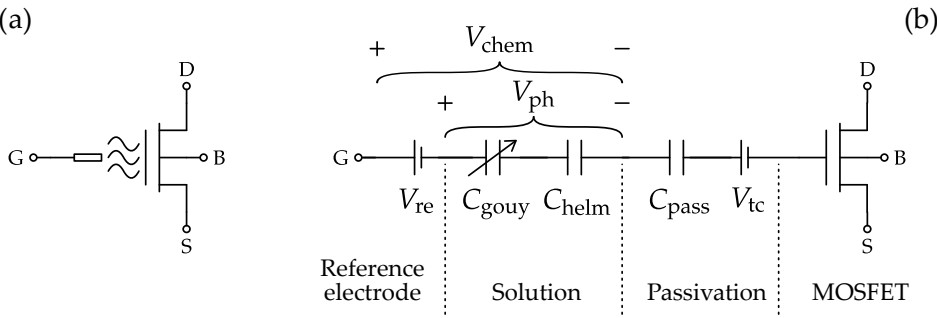

**Figure 19.** Ion-selective field-effect transistor (ISFET) symbol (**a**) and equivalent electrical model (**b**).

The corresponding pH-dependent shift on the equivalent ISFET threshold voltage can be expressed as:

$$\Delta V_{\text{TH}} = \gamma + \alpha S_{\text{N}} \text{ pH},\tag{26}$$

where $\gamma$ is a chemical constant, $S_{\text{N}}$ stands for the theoretical Nernstian pH sensitivity (typ. 59 mV/pH) and $\alpha$ counts for the sensitivity deviations due to the double-layer capacitance [169].

Several specific AFE circuits have been proposed for the ISFET from the very beginning of its birth. However, the last decade has experienced an exponential growth in interface topologies thanks to the emergence of integrated arrays of ISFETs for lab-on-a-chip instruments [170] with pixel-like CMOS read-out circuits monolithically attached one-by-one to their corresponding ISFET sensors. In this review, two types of ISFET interfaces are presented according to potentiometric and amperometric read-out methods.

Figure 20 shows a summary of the common ISFET potentiometric configurations, where $V_{\text{ref}}$, $V_{\text{A}}$ and $d_{\text{o}}$ stand for the RE potential, the AFE output and the digital output of the overall smart ISFET front-end, respectively. The most classic solution is probably the constant-$I_{\text{D}}$ constant-$V_{\text{DS}}$ dual-OpAmp topology [171] of Figure 20a due to its excellent linearity in all regions of MOSFET operation, from weak to strong inversion and from conduction to saturation. Its principle of operation can be explained as follows: the current sink $I_2$ ensures constant-$I_{\text{D}}$ for the ISFET device $M_1$, which is configured here as a gate-to-source voltage follower; the unity-gain OpAmp $A_2$ provides the low-impedance copy $V_A$ of the ISFET source voltage for further A/D conversion; $I_1$ and $R_1$ define the selected constant $V_{\text{DS}}$ bias, which is then applied to $M_1$ by the unity-gain OpAmp $A_1$. Based on the same strategy, a single-OpAmp topology is presented in Figure 20b. In both cases, a separated substrate (i.e., local well) for the ISFET is needed in order to avoid any CMOS technology dependency. For example, if the floating-gate MOSFET $M_1$ is biased in strong inversion forward saturation, the EKV model [172] returns the following large-signal DC characteristic:

$$I_{\text{D}} = \frac{\beta}{2n} \left(V_{\text{GB}} - V_{\text{TH}} - n V_{\text{SB}}\right)^2 \left(1 + \lambda V_{\text{DS}}\right),\tag{27}$$

where $\beta$ is the current factor (including device aspect ratio), $n$ is the subthreshold slope and $\lambda$ is the channel-length modulation parameter. Clearly, any variation of the gate voltage $\Delta V_{\text{GB}}$ (or alternatively of the threshold voltage $\Delta V_{\text{TH}}$) in Equation (27) due to pH will cause, at constant $I_{\text{D}}$ and $V_{\text{DS}}$, the corresponding change in the source voltage $\Delta V_{\text{SB}} = \Delta V_{\text{TH}}/n$. This subthreshold slope dependency can be avoided by forcing $V_{\text{SB}} = 0$ (i.e., local well). When this technology option is not available, like in twin-well CMOS processes, the alternative topology [173] of Figure 20c is more suitable. In this case, a matched MOSFET $M_2$ is introduced to form a differential pair with ISFET $M_1$. If both devices are biased at the same constant current (i.e., $I_1 = I_2 = I_3/2$), the high-gain negative feedback will force both devices to share the same $V_{\text{DS}}$ and consequently, from Equation (27), $M_2$ will follow any voltage variation at the floating gate of $M_1$ even under substrate effects ($V_{\text{SB}} \neq 0$).

The common approach of the ISFET interface circuits presented in Figure 20 is the usage of the floating-gate MOS transistor as a voltage follower. However, the true nature of the MOSFET is of a transconductor, which is a voltage-controlled current source, which opens up the possibility for ISFET amperometric topologies. Compared to potentiometric techniques, current-mode read out can enjoy those circuit design techniques described in Section 4.2. In addition, signals in the current domain can be directly filtered or employed in modulation schemes with the inclusion of CMOS integrators. On the other hand, ISFET amperometric circuit strategies tend to suffer from the nonlinear $I/V$ MOS device characteristics.

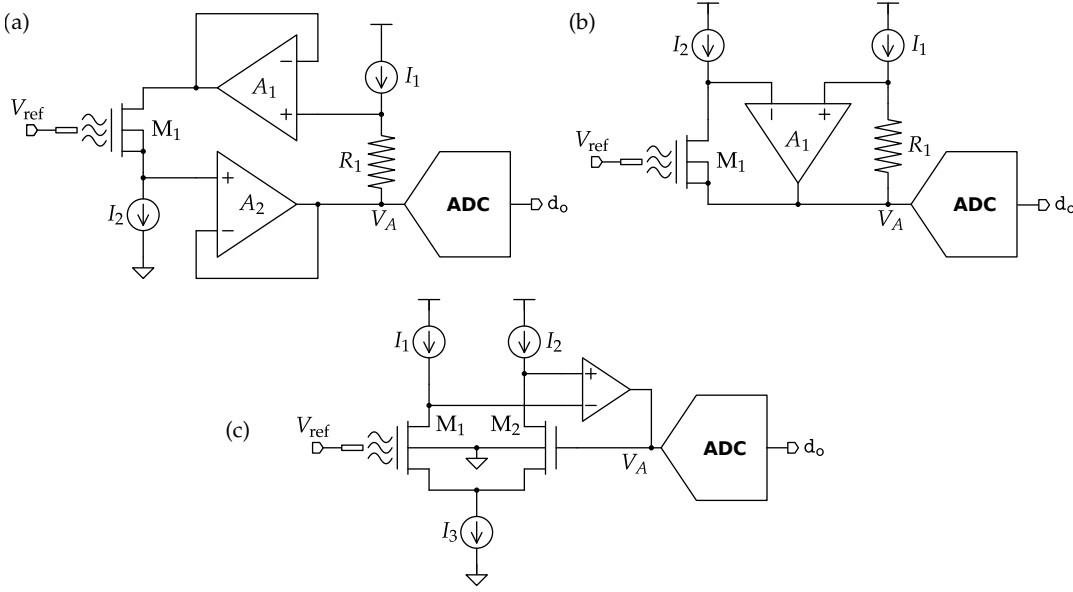

**Figure 20.** Potentiometric ISFET interfaces: (**a**) constant-$I_D$ constant-$V_{DS}$ classic dual-OpAmp and (**b**) single-OpAmp topologies, and (**c**) constant-$I_D$ OpAmp follower.

Figure 21 presents two common ISFET amperometric interfaces. In both examples, the floating-gate MOSFET is biased in its strong inversion triode region by selecting $V_{bias} \leq V_{Dsat}$. According to the EKV model [172],

$$I_D = \beta \left[ V_{GB} - V_{TH} - \frac{n}{2} \left( V_{DB} + V_{SB} \right) \right] V_{DS}. \tag{28}$$

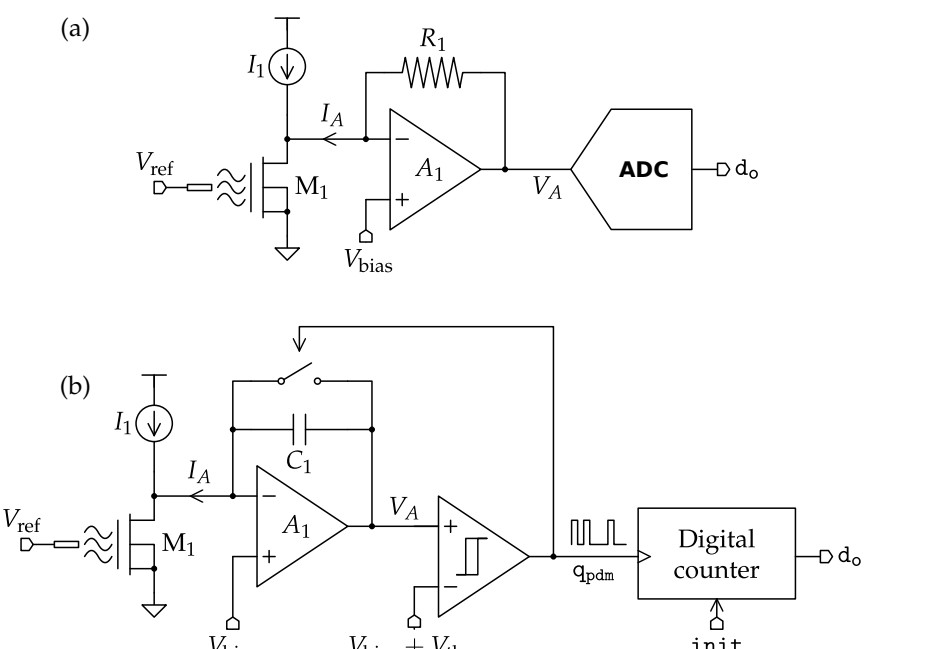

**Figure 21.** Amperometric ISFET readout interfaces: (**a**) classic resistive TIA analog front-end and (**b**) asynchronous pulse-density modulation (PDM) analog-to-digital converter with capacitive-TIA.

Thus, the current-mode signal $I_A$ shows a linear behaviour respect to any change in the floating-gate voltage of the ISFET $M_1$ with a transconductance gain $\beta V_{bias}$. In the classic topology of Figure 21a, $I_A$ is converted back to the voltage domain $V_A$ through the resistive TIA $R_1 A_1$ for further A/D conversion [174]. A completely different approach is followed in the interface of Figure 21b, where the same current-domain signal is directly employed for the A/D conversion. Its principle

of operation can be summarized as follows: $I_A$ is integrated through the CTIA $C_1 A_1$; the resulting signal $V_A$ is then 1-bit quantized at $q_{pdm}$ by the comparator with hysteresis according to the voltage threshold $V_{th}$; the same digital signal is fed back to the CTIA as the trigger to reset the continuous-time analog integration. As a result, $q_{pdm}$ shows a pulse density modulation (PDM) with respect to $I_A$ (pH). The only remaining signal processing step to complete the A/D conversion of $I_A$ is some sort of digital low-pass filtering, which is implemented in Figure 21b by a simple digital counter playing the role of a first-order low-pass filter. Apart from this asynchronous integrated-and-fire strategy, other $I_A$ modulation schemes are also possible for the practical implementation of the A/D conversion, like pulse width modulation (PWM) [175].

### 4.4. Impedance Spectroscopy Interfaces

An EIS circuit must be capable of measuring how the complex small-signal impedance of an electrolyte/electrode interface varies with frequency. The aim is determining the main features of the typical curve shown in Figure 5 (right). Simpler systems designed to measure only the capacitive component at fixed frequencies are sometimes used for their simplicity when the goal is finding the coverage of the functionalized working electrode by the target molecules [176] or detecting cellular activity [177]. Classification of the EIS systems proposed in the literature can be made on the basis of the diagram of Figure 22.

The stimulus generator produces a reference waveform, which is applied to the EIS interface through a proper electrode configuration. The response is then processed to extract the complex impedance (real and imaginary part). Two main working principles are possible: the fast Fourier transform (FFT) and the frequency response analysis (FRA). The former consists of stimulating the EIS interface with a waveform having a broad spectrum, such as a single pulse or a square waveform. The response is converted to digital and FFT is performed in order to extract the required dependence of the impedance on frequency. The FFT approach is not the optimal solution for compact, single-chip EIS systems, due to the requirement of complex digital-signal processing (DSP) functionalities.

With the FRA approach, the stimulus is sinusoidal and the response is analyzed in order to extract the complex impedance at the stimulus frequency. The impedance curve is then obtained through a frequency scan over the interval of interest. In both cases, the AC stimulus has a small magnitude (tens of mV) and it must be superimposed on a larger DC component, which is used to activate the desired redox processes. When the stimulus is applied as a voltage, imposed between the reference electrode and the working electrode, the current flowing through the working electrode is acquired. Current stimulation is not common in EIS systems but is frequently used in bio-impedance measurement devices [178–180], where it is not important to set a precise DC bias.

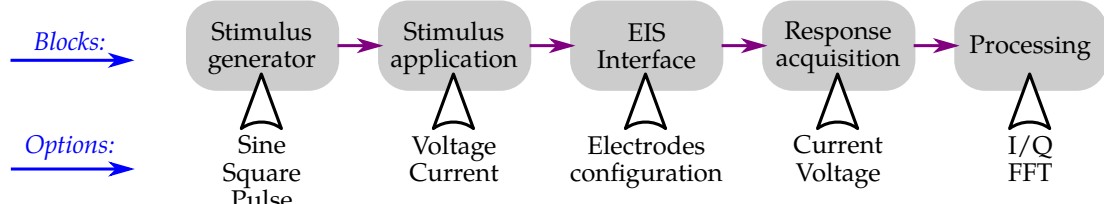

**Figure 22.** Fundamental elements of an electrochemical impedance spectroscopy (EIS) interface.

The most popular approach for the development of compact EIS systems is using voltage stimulation combined with FRA operating principles. Accurate EIS approaches involve using the typical three-electrode potentiostat configuration (three-electrode connection), where the voltage stimulus is imposed across the RE and WEs through the action of a CE, as in voltammetry systems. However, due to its simplicity, the two-electrode is preferred for large sensor arrays where a large common CE is coupled with an array of smaller WEs (usually microelectrodes) [181,182]. In such experiments, the current levels at WEs are low enough that, even if summed up at the large common CE, its potential drift is negligible and a sufficiently precise potentiostatic control is established.

FRA requires a sinusoidal stimulus; square wave AC stimulation is generally considered not adequate due to the excessively high content of harmonics that would alter the measurement [181,183]. Due to the need for sweeping the frequency across intervals several decades wide, classical RC or LC oscillators are not adequate for generating the stimulus. The preferred solution is synthesizing the sinusoid by means of a conventional *R*-2*R* [181] or resistor-string [179] DAC, starting from a set of samples.

High spectral purity requires high resolution DACs and a large number of samples per cycle, which are factors that lead to an increase of power consumption and architectural complexity. An alternative approach is starting from a square waveform and filtering it to select only the fundamental component. Due to the wide frequency interval, time-continuous filtering is not feasible. In [184] synchronization of the filter-bandwidth with the stimulus frequency is obtained by using a switched capacitors 5th order Chebyshev type-I filter clocked at a multiple of the input frequency.

The FRA approach requires that the response of the interface under test (IUT) be processed to extract the impedance real and imaginary parts. The IUT response is either a voltage or a current. The straightforward solution is shown in Figure 23, where the IUT response is multiplied by a stimulus time reference (STR), i.e., a signal synchronous with the input stimulus. Multiplication for the STR and for the STR delayed by one-fourth of period ($\pi/2$ phase delay) produces two signals whose DC components are proportional to the in-phase (I) and quadrature (Q) components of the IUT response, respectively. For a voltage stimulus and current response, the I and Q signals are proportional to the real and imaginary parts of the electrochemical cell admittance.

The scheme of Figure 23 is frequently implemented by all-analog circuits, where the multipliers are formed by switch matrices and the STR is a square waveform. Even with this type of simplification, full integration of the scheme in Figure 23 can be quite challenging, due to the difficulty of designing low pass filters (LPF) with sub-Hz cut-off frequencies using the pF-range capacitors available on-chip. In Ref. [179], a Gm-C filter that relies on external capacitors is employed, whereas an external filter/amplifier is used in [182].

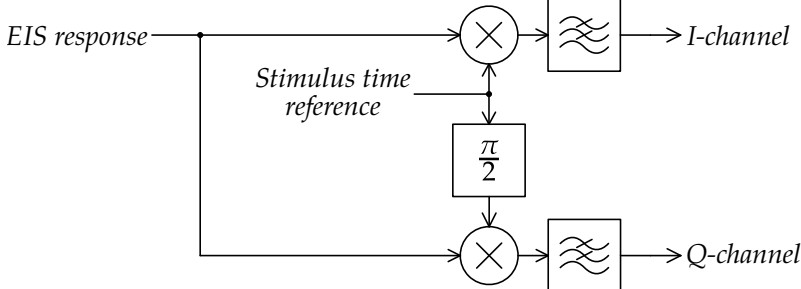

**Figure 23.** I/Q demodulation of the EIS response.

An alternative solution, corresponding to multiplying the EIS interface response by a discretized sinusoidal reference, is proposed in [181]. Such a result is obtained by digitizing the response by means of a dual-slope ADC, whose integration time is digitally modulated by the sinusoidal STR. An advantage of this solution is that a digital signal is produced and no further filtering is required. Another mixed analog-digital approach based on a pass-band delta-sigma ADC is sketched in [185], while all-digital demodulation solutions can be obtained by processing the digitized EIS interface response in the digital domain to calculate the single-frequency the Discrete Fourier Transform (DFT). Such a solution [186] represents an FFT/FRA hybrid approach and allows for relaxed DSP requirements with respect to pure FFT based approaches, but sinusoidal excitation and the execution of a frequency scan are necessary as in FRA. As an alternative to extracting the I and Q components, separate magnitude and phase estimation has been demonstrated to be feasible [187].

### 4.5. Readout Interfaces for Bio-Fuel Cells

Amongst the latest trends, sensing based on biological fuel cells (BFCs) is a promising approach for wearables. Different definitions of a BFC may apply [188]; however, in the present context, a BFC can be defined as a device that utilizes at least one biogenic catalyst, such as enzymes, microbes or microorganisms, in order to assimilate a fuel and an oxidant to generate electric power. BFCs are able to generate charge and simultaneously store it in large double layer and/or pseudo-capacitance structures around the electrodes. In a BFC, as in any generic fuel cell, the amount of available power $P_{out}$ is set by the concentration of the fuel. At the same time, a BFC based on glucose oxidase (GOx) or lactate oxidase (LOx) may use respectively glucose or lactate, naturally present in sweat, as fuels [189]. In turn, since glucose and lactate are markers of metabolic activity, they can be leveraged to provide both information and energy for information readout, digitalization and transmission, giving yield to a fully self-powered smart wearable. Power densities in the order of $1 \text{ mW/cm}^2$ have been recently demonstrated for on-body lactate BFCs [78,190].

From the electrical point of view, a BFC behaves as a voltage source $V_{cell}$ able to sustain a certain amount of current $I_{cell}$, whose intensity depends on the concentration of the fuel. Schematic polarization curves are shown in Figure 24. For an unloaded BFC, i.e., in the open-circuit condition, $V_{cell}$ is at a maximum and it is usually on the order of 0.3–0.5 V. Once $I_{cell}$ starts to be provided to a load, $V_{cell}$ drops in a resistive-like manner up to the overloading condition of the BFC, where $V_{cell}$ drops dramatically to zero. The power versus cell voltage characteristic reveals a maximum power point (MPP) which optimizes the power harvested from the BFC. However from the CMOS point of view, operation at MPP, which falls around $V_{cell} \approx 0.2 \text{ V}$, is extremely challenging and different strategies may be adopted.

A possible solution is depicted in Figure 25a [191]. This system uses energy supplied by the BFC and provides glucose/lactate concentration readout and RF data-transmission in the UHF band. The system operates in two phases: during $\varphi_{charge}$, the BFC is charging an external ceramic capacitor $C_{ext}$, employed as auxiliary energy storage element. Provided that $\varphi_{charge}$ is long enough, the transient currents finally approach zero, meaning that the available supply voltage for circuit operation will be the open circuit value of $V_{cell}$. During $\varphi_{readout}$, the circuit is turned on and supplied by the harvested energy in $C_{ext}$. In this phase, an approximated matched load $R_{MPP}$ is connected to the BFC in order to track the MPP. This need is justified by the authors in [191] by experimental evidence disclosing a linear relationship between the $P_{out}$ and analyte concentration when operating at MPP.

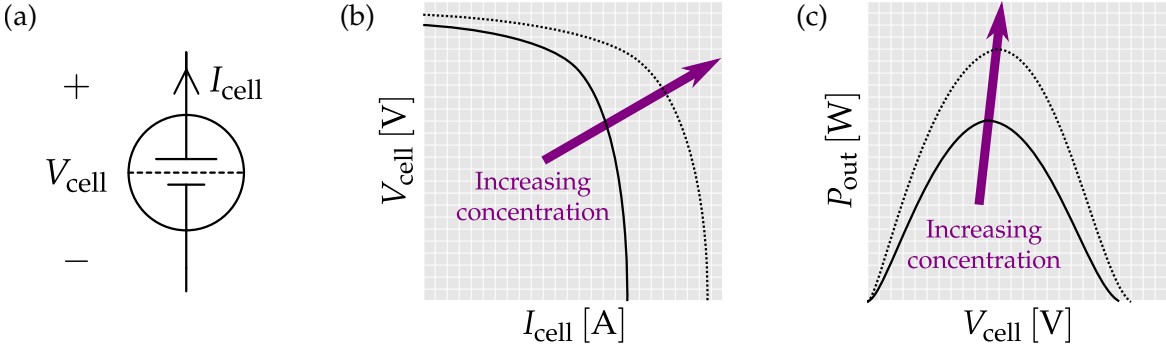

**Figure 24.** Bio-fuel cell symbol (**a**) and schematic polarization curves: $V_{cell}$ versus $I_{cell}$ (**b**) and output power $P_{out}$ versus $V_{cell}$ (**c**).

A more compact solution for BFC interfacing is proposed in [192] and it is reported in Figure 25b. Here, an all-digital approach is followed comprising a supply-controlled ring oscillator, a pulse shaper and an impulse-based transmitter for inductive-coupling. Linearity performances are worse with respect to the previous solution; however, only three connections to external components are required, with clear advantages to the heterogeneous integration of the chip onto a flexible substrate. Using the zero-$V$th CMOS transistors, the test chip is able to function with $V_{cell}$ as low as 0.23 V.

While both reviewed works present pioneer solutions, it is worth noting that [191] fails to address the physiological normal ranges of Table 1. On the other hand, Ref. [192] is interfaced to a BFC fuelled by fructose, which is an analyte not present in human body fluids. Thus, BFC interfacing is still a green field and many more solutions are expected to be proposed over the next years.

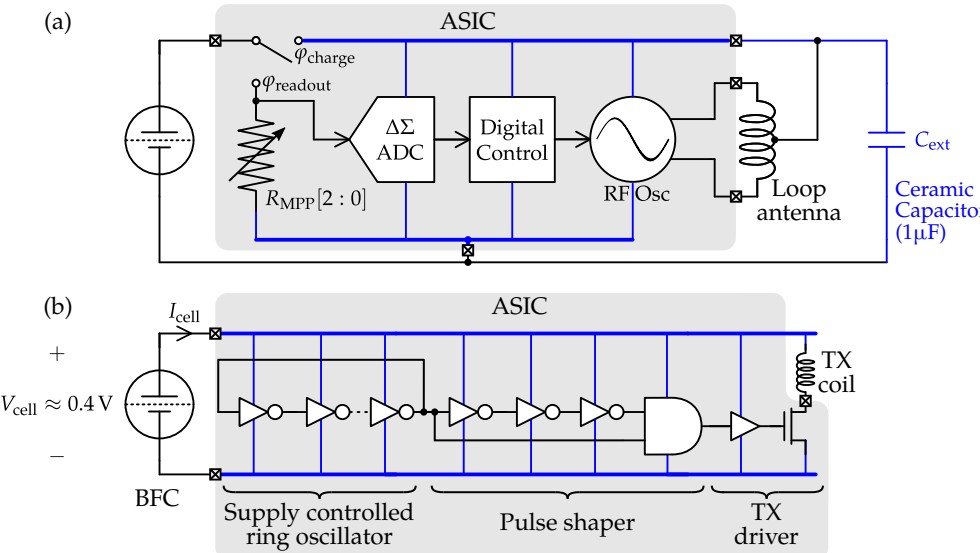

**Figure 25.** CMOS interface for self-powered fuel cell sensors. (a) two-phase operation approach; (b) free-running oscillator approach.

## 5. Conclusions

We have reviewed the state-of-the-art CMOS interfaces for non-invasive monitoring of physiological parameters through electro-analytical techniques in body fluids. Such integrated circuits constitute the first stage of the signal acquisition and processing chain in a system comprising (bio)electrochemical sensors to be embedded in mechanically flexible substrates. While constrained to be low-power and to have a small footprint, these front-end circuits bridge the gap between the transduced low-level signals and Cloud-connected components such as smartphones, smartwatches or dedicated WPAN links. We coined the term IoW to identify these kinds of smart systems that find application in health, sports, safety at work, defence and law enforcement. With the aim to provide a self-contained framework, an overview of electro-analytical methods including potentiometric, amperometric and impedimetric sensing has been given. Challenges in CMOS circuit design for potentiometric sensing, both by the means of ISE and ISFET sensing elements, have been stated and analysed. CMOS potentiostats for amperometric sensing, popular thanks to its customization to a vast amount of target analytes, have been extensively discussed, comprising the most recent compact solutions. More complex systems such as those based on EIS need sophisticated analog, mixed-signal and digital signal processing. This implies a penalty in terms of silicon area and power consumption directly impacting wearable cost and portability. Finally, BFC-based smart systems, still in their embryonic stage, have been identified as a promising emerging field thanks to their self-powering characteristics.

**Author Contributions:** Conceptualization, M.D. and F.J.d.C.; Methodology, M.D.; Supervision, F.J.d.C., P.B., F.S.-G.; Writing and Editing, all authors.

**Funding:** The research leading to these results has received funding from the People Programme (Marie Curie Actions) of the Seventh Framework Programme of the European Union (FP7/2007–2013) under REA Grant No. 600388 (TECNIOspring programme: TECSPR16-1-0056), and from the Agency for Business Competitiveness of the Government of Catalonia, ACCIÓ. It has also received funding from "PRODUCTE 2016" grants (PROD-00114) from the knowledge industry program run by the Universities and Research secretariat from the Catalan regional government.

**Conflicts of Interest:** The authors declare no conflict of interest. The founding sponsors had no role in the design of the study; in the collection, analyses, or interpretation of data; in the writing of the manuscript, or in the decision to publish the results.

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
