# Peer review of "CMOS Interfaces for Internet-of-Wearables Electrochemical Sensors: Trends and Challenges"

_electronics, doi:10.3390/electronics8020150_

Round 1

Reviewer 1 Report

In this review paper, the fundamentals of electrochemical sensing mainly used in the field of wearables with the related CMOS interfaces applied for electrochemical sensors, as well as the latest trends in smart wearables is described. It is a very well organized and nicely written manuscript and of great importance. I believe this manuscript can be accepted for publications in Electronics but just minor revisions are necessary before publishing. 

Still more recent relevant references need to be added, please see my comments below for more details.

1)      In section 2.1. “Wearables and non-invasive monitoring”, authors should add the very recent review paper on wearable electronics (Wearable Bioelectronics: Enzyme-Based Body-Worn Electronic Devices, Acc. Chem. Res., 2018, 51 (11), pp 2820–2828).

2)      In the same section, more applications (security and environmental) of wearable electrochemical sensors are not covered. Authors can refer to the following references and cite them too.

-          https://doi.org/10.1016/j.snb.2018.07.001 (Sensors and Actuators B: Chemical 273, 966-972, 2018)

-          DOI: 10.1021/acssensors.7b00051 (ACS Sens.20172 (4), pp 553–561)

-          https://doi.org/10.1016/j.bios.2017.10.044 (Biosensors and Bioelectronics 101, 227-234, 2018)

3)       Also the below recent papers for sweat analysis should be added too:

-          https://doi.org/10.1016/j.bios.2016.09.038

-          https://doi.org/10.1002/advs.201800880

-          DOI: 10.1038/srep23111

-          https://doi.org/10.1002/elan.201800414

-          https://doi.org/10.1002/smll.201802876

-           

4)      The following reference can be added to section 4.1. Potentiometric Interfaces.

-          DOI: 10.1109/JSSC.2018.2815657 (IEEE Journal of Solid-State Circuits)

5)      Reference 78, 112, 136, the year is not Bold.

6)      Basically, I recommend double checking the reference format again.

Author Response

Dear reviewer, please find attached the detailed response in the pdf.

Kind regards.

Reviewer 2 Report

The authors provide a survey of CMOS-based interfacing approaches for electrochemical sensors, i.e. read-out circuits for the measurement electrodes. Special focus is put on applications in wearables. First, a short overview is given of typical application scenarios, the analytes and the body fluids that these measurement quantities are extracted from. The authors then describe the basics of the three fundamental measurement approaches, in order to cover state-of-the-art circuitry approaches for the realization of these measurement approaches in the fourth and main section of the manuscript. Here, amperometric interfaces have the largest weight. The manuscript ends with a short conclusion.

The manuscript is well written with a clear structure. It was a good idea to give an overview of the basic measurement principles and then to discuss their different realizations. The information given is well illustrated by schematics, plots etc. Formulae are cited, while lengthy derivation are left to the literature.

The focus of the manuscript is clearly put on the circuitry concepts, with their pros and cons, particularly with regards to wearability, being discussed. The aspect of connectivity, in contrast, is only touched in 1-page tour through smartphones and smartwatches as well as wireless communication technologies. This weighting was definitely a good choice and benefits the main topic; however, I wonder why the authors do not simply talk about wearables instead of IoW devices. The coining of a new term, with one of the key aspects hardly addressed in the paper, is a bit overdone.

Section 2, the overview of smart wearables and the IoT, is rather a motivation than a systematic overview. This becomes particularly clear in section 2.2, "Wearable sensors in health": To summarize the activities in this field, reference is given to a paper from 1992, which describes wearables in one single column. Then, the authors solely discuss diabetes. Similarly, section 2.3 concentrate on sweat right at the beginning and just mentions blood lactate measurement, one of the classical methods in sports, shortly at the end. The remaining subsection similarly have an examplary rather than systematic character.

A few minor recommendations:
In the introduction of section 2, when physical and biochemical sensors are contrasted, the particularities of the latter should be indicated.
Please check the quantities in the formulae: Not all of them are explained in the text but should be so for the sake of completeness.
Also check the references: The author names, journal volume and number as well as page number are differently formatted, journal/conference names are partly given in full, partly abbreviated.

Author Response

(The authors gave the same response as above.)
